# Ocean acidification disrupts the biomineralization process in the oyster *Crassostrea virginica* via intracellular calcium signaling dysregulation

Chi Huang[1], Joseph Matt[2,3], Christopher Hollenbeck[2,3], Leisha Martin[2] & Wei Xu ✪ [1] ✉

Calcium is a key component in the shell and skeleton structure, serving as a second messenger for regulating biomineralization across many species. Ocean acidification (OA) is well-studied for causing shell dissolution in marine bivalve species by disordering calcium deposition. However, the regulatory pathway of calcification affected by OA remains unclear. This study assessed eastern oyster (*Crassostrea virginica*) to determine how calcium signaling responds to elevated $pCO_2$ and influences shell formation. Under elevated $pCO_2$, increased calcium influx was found in mantle epithelial cells, followed by the upregulation of calmodulin, a primary sensor of intracellular calcium. Expression levels of shell matrix proteins (SMPs), representing shell construction conditions, were significantly upregulated in the $CO_2$-induced mantle cells. Larval *C. virginica* exhibited developmental stage-dependent alterations in calcium signaling and SMPs disarrangement stimulated by $pCO_2$. Pharmaceutical blockage of the calcium binding on calmodulin induced abnormal expression of downstream genes and shell matrix changes consistent with those caused by elevated $pCO_2$. Restored SMPs expressions in $CO_2$-treated mantle cells were achieved by rescuing the level of calcineurin, a downstream effector of calmodulin. These findings suggest that shell deformities under OA are primarily caused by the disruption of the calcium-calmodulin signaling pathway in mantle epithelial cells.

Biomineralization in marine calcifiers, such as mollusks, corals, and echinoderms, form shells or skeletons through their ability to precipitate calcium carbonate $(CaCO_3)$[1,2]. As one of the major calcifier groups, marine bivalves have important ecological functions and commercial value worldwide[3,4]. Biomineralization is fundamental to marine bivalve shell formation, which serves as a physical barrier against predators, environmental stressors, and desiccation. The shells of bivalves are composed of 95% $CaCO_3$ and 1–5% organic matrix[5,6]. Previous studies have shown that the shell formation process is initiated by mantle tissues, where the $CaCO_3$ is secreted to facilitate shell growth[7,8]. The mantle epithelial cells produce components of the shell organic matrix, which regulate the nucleation, regulation, and structural organization of $CaCO_3$[9,10].

Over the past decades, it has been speculated that a cascade of genetic regulation controls the shell formation process in marine bivalves[10–14]. More than 250 shell formation-related genes are predicted from the genome of *Magallana gigas* (previously *Crassostrea gigas*)[15,16]. In particular, shell matrix protein (SMP) production and ion transportation pathways are suggested to be key genetic processes in regulating biomineralization of bivalves[11,17–21]. Several signaling pathways, including Wnt[22], TGF-β/BMP[23], and a subunit of guanine nucleotide-binding protein (G-protein)[24] are also implicated in organizing shell formation process in various marine bivalve species. In addition, some transcription factors, including homeobox genes, NF-κB, zinc finger proteins, and Smad family proteins, are identified as regulators of SMPs in marine bivalve[25–29]. However, the precise genetic pathways controlling shell formation remain unclear.

Calcium is not only an essential component of biomineralizing organisms but also serves as a crucial intracellular second messenger regulating numerous signaling pathways. Many studies have reported that

[1]Department of Veterinary Physiology and Pharmacology, College of Veterinary Medicine & Biomedical Science, Texas A&M University, College Station, TX, USA. [2]Department of Life Sciences, College of Science, Texas A&M University - Corpus Christi, Corpus Christi, TX, USA. [3]Corpus Christi Research and Extension Center, Texas A&M AgriLife Research, Corpus Christi, TX, USA. ✉e-mail: wxu1@tamu.edu

calcium signaling-related genes share a high degree of conservation throughout evolution between mollusks and mammals[30,31]. Calmodulin (CaM), a primary calcium sensor in calcium-signaling pathways, has been identified in various bivalve species[31–34], and mantle tissue related to the shell formation process shows the highest expression of CaM[32]. CaM and CaM-like proteins are reported to be present in the bivalve shell layer, and have been implicated in aragonite nucleation in conjunction with other shell organic matrix[18,35]. As a calcium/CaM-dependent phosphatase, calcineurin (CaN) is directly activated upon binding of the calcium-CaM complex[36,37]. CaN is also detected mainly localized in the inner epithelial cells of the mantle[30], and is reported to play a role in regulating the shell formation process in *P. fucata*[38]. This process is homologous to the role of CaN in bone formation and absorption processes in vertebrate species[39–41].

Anthropogenic activities significantly increase atmospheric $CO_2$ levels, with projections indicating a rise from 400 ppm to 1000 ppm by the year 2100[42]. The ocean absorbs ~30% of the rapidly rising atmospheric $CO_2$, leading to ocean acidification (OA)[43–45]. The dissolution of $CO_2$ in seawater results in altering pH levels and carbonate system in the ocean[46]. There is strong evidence that marine organisms are suffering from increased acidification[47–49]. Compared with vertebrate species, OA negatively affects more biological responses in invertebrate species, especially in calcification and biomechanics. The shifts of pH and carbonate system in the ocean ecosystem disrupt biomineralization processes in marine calcifiers by the dissolution of calcareous structures $(CO_2 + H_2O + CaCO_3 = > 2HCO_3^- + Ca^{2+})$[43,50,51] and reduction of carbonate ions for $CaCO_3$ precipitation $(CO_2 + H_2O = > HCO_3^- + H^+;$ $H^+ + CO_3^{2-} = > HCO_3^-)$[43,52,53].

While the chemical impacts of OA on bivalve calcification are comprehensively described, it remains unclear whether bivalve shell formation also responds physiologically to OA conditions. Bivalve shell mineralization occurs within the extrapallial fluid (EPF), a compartment located between the shell and mantle tissue, where pH and carbonate chemistry parameters differ markedly from those in seawater[54]. A previous OA simulation study shows that EPF pH and other carbonate system parameters exhibit different modifications from those in the surrounding seawater, reflecting a self-regulatory system in the bivalves[55]. These findings suggest that mantle epithelial cells, the major shell-forming cell population directly exposed to the EPF, may exhibit physiological responses to the altered conditions under OA.

In addition, there is strong evidence that OA poses a threat to the shell formation process in marine bivalves by disrupting calcium absorption and deposition in mantle tissue[1,56–59]. The fluctuations of intracellular calcium and transcriptional changes in calcium-signaling-related genes in the gill, mantle, or hemocytes of various marine bivalve species are suggested as a compensatory mechanism to maintain calcium deposition for shell formation under the stress of OA[34,60–63]. Several studies have indicated that simulated OA conditions may disrupt cellular signaling pathways associated with immune responses in marine organisms by altering intracellular calcium concentration in hemocytes[64–66]. Changes in extracellular pH have been reported to alter proton gradients, membrane potential, and the activities of several ion channels in oyster mantle epithelial cells[67]. Additionally, OA-induced fluctuations in carbonate chemistry and extracellular $Ca^{2+}$ may influence $Ca^{2+}$ channel activity and disrupt intracellular $Ca^{2+}$ level, as reported in oyster hemocytes[61,63]. Therefore, there is a plausible hypothesis that OA conditions may disturb the intracellular calcium homeostasis and dysregulate the genetic mechanism of the calcium-signaling pathway, ultimately impairing the bivalve shell formation.

Due to the important characteristics of calcium for numerous intracellular processes and as the major component of bivalve shells, the role of intracellular calcium in biomineralization under OA conditions was explored in this study. We examined how elevated $pCO_2$ affected bivalve biomineralization through the calcium-signaling pathway by establishing an in vitro epithelial cell-dominated model using primary mantle cell culture from the eastern oyster, *Crassostrea virginica*. An in vivo analysis of biomineralization-related response to OA induced by elevated $pCO_2$ across early developmental stages of *C. virginica* was also conducted to elucidate the role of calcium signaling in early shell formation. Our findings suggest

that OA-induced shell deformation in marine bivalves is associated with the dysregulation of intracellular calcium-signaling pathway.

## Results

### Intracellular calcium influx in *C. virginica* mantle cells under elevated $CO_2$

To investigate the impact of OA on the cellular functions in biomineralization-related organs, the mantle tissues of *C. virginica* were dissected for primary cell culture (Fig. 1A). Various cell types, including epithelial-like cells, hemocyte-like cells, and fibroblast-like cells, migrated from the explant within 48 h (Fig. 1B). The mantle cells from non-adhered explants or the cell dissociation were detached within 24 h, whereas the cells that migrated from adhered explants remained viable for over two weeks. Hemocyte-like cells were progressively diminished from the adherent cell populations following culture medium replacement after 3–4 days. Less than 10% of fibroblast-like cells were detected in the migrated cell populations around each collagenase-treated explant (Fig. 1B and Supplementary Fig. 3C). After 14 days, the epithelial-like cells migrated around the explants, forming ~70% confluence in the surrounding areas and becoming the dominant cell type (Fig. 1C, D and Supplementary Fig. 3C). While most epithelial-like cells degenerated within two weeks, some remained viable for up to a month.

To determine whether the elevated $pCO_2$ condition (simulated OA) can dysregulate calcium-signaling pathways in the oyster mantle cells, the intracellular calcium level was monitored in $CO_2$-treated mantle cells using calcium indicator Fluo4-AM. Following a short-term $CO_2$ treatment (1 h), the average fluorescent intensity of the calcium signal in mantle cells significantly increased by 2.2% compared to their pre-exposure levels (Fig. 1E, F, $p = 0.01567$). Additionally, flow cytometry analysis showed that the average fluorescence intensity of calcium in the mantle cells under long-term $CO_2$ exposure (24 h) was 82.17%, significantly higher than the mantle cells in the control group (Fig. 1G, H, $p = 0.03887$).

### Dysregulation of calcium-signaling pathway and SMPs in *C. virginica* mantle cells under elevated $pCO_2$

Two calcium-signaling-related proteins, CaM (*Cv-CaM*) and CaN (*Cv-CaN*), were assessed to evaluate the cellular responses of calcium-signaling pathway to increased $pCO_2$ level in oyster mantle cells. Expression patterns of the *Cv-CaM* were significantly upregulated at the mRNA level by 2.47± 0.69-fold ($p = 0.008735$) under 1.5% $CO_2$ condition compared with mantle cells with an ambient $CO_2$ environment (Fig. 2A). However, *Cv-CaN*, the downstream gene of *Cv-CaM*, showed a contrasting pattern of mRNA expression. The expression of *Cv-CaN* was downregulated significantly to 0.75± 0.19-fold ($p = 0.0471$) in 1.5% $CO_2$-treated mantle cells compared to the control group (Fig. 2B). Similarly, the protein expression of CaM and CaN quantified by the immunofluorescence (IF) analysis showed a pattern consistent with the transcriptional analysis. The CaM expression at the protein level increased by 112.53% in the *C. virginica* mantle cells cultured in 1.5% atmospheric $CO_2$ exposure compared to the cells in the control group (Fig. 2C, $p = 0.0042$). The *C. virginica* mantle cells dramatically decreased production of CaN protein by 86.46% under the 1.5% $CO_2$ condition (Fig. 2D, $p = 0.023$).

In the mantle epithelial cells, the transcription of SMPs, which are well-studied for their role in mollusks shell formation, were utilized to assess the responses of shell formation to elevated $CO_2$ concentration. Despite species-specific variations in SMPs, SMP repertoires are reported to share four functional domains, including von Willebrand factor type A domains (VWA), chitin-binding-2 domains (CB-2), carbonic anhydrase domains (CA), and tyrosinase domains in diverse molluscan species[68,69]. The presence of these shared domains in different SMPs suggests a conserved biomineralization toolkit across bivalve species. Therefore, we selected four conserved SMPs-encoding genes to represent the shell construction conditions at the molecular level, including Pif97 (*Cv-Pif97*, VWA and CB-2 domains)[13], Tyrosinase (*Cv-Tyr*, Tyrosinase domain)[70], Nacrein (*Cv-Nacrein*, CA domain)[71,72], and Chitin synthase (*Cv-Chit*)[73,74]. The amino sequences of these corresponding proteins in *C. virginica* showed similarly

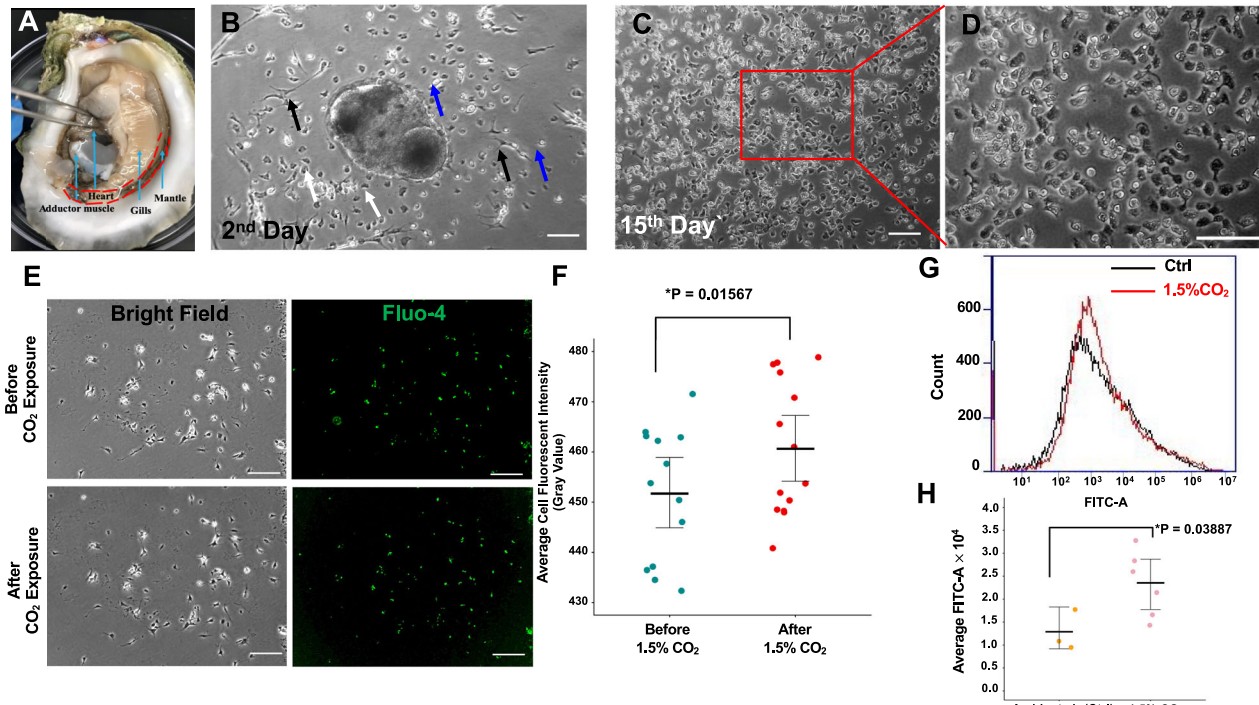

**Fig. 1 | Calcium flux in *C. virginia* mantle cells under elevated $CO_2$ stress. A** Soft body of the eastern oyster *C. virginica*. The muscle, heart, gill, and mantle were blue arrowed, respectively. The dissection location of the mantle tissue for cell isolation was indicated by a red dashed line. **B** *C. virginica* mantle explant and isolated mantle cells after 48-h seeding. White arrows = epithelial-like cells; blue arrows = hemocyte-like cells; black arrows = fibroblast-like cells. **C** *C. virginica* mantle cells cultured on day 15. **D** A magnified view of C showed a majority of epithelial-like cells. **E** Loading of Fluo-4/AM in the isolated mantle epithelial cells before and after 1.5% $CO_2$ short-term exposure. **F** Comparison of labeled calcium signal intensities in the *C. virginica* mantle cells before and after 1.5% $CO_2$ 1-h short-term exposure ($n = 5$ images per oyster). **G** Fluorescence intensity of intracellular calcium in *C. virginica* mantle cells recorded by a flow cytometer between the control condition and the 24-h long-term $CO_2$ exposure. **H** Mean fluorescence intensity of intracellular calcium level ($n = 3$–6 oysters). *$P < 0.05$, statistical significance between the control and $CO_2$-treated group was determined using a Student's *t* test. White scale bar: 100 μm. Error bar: 95% confidence interval of the mean.

conserved domain regions from different bivalve species, including *Pinctada fucata*, *Magallana gigas*, *Mytilus edulis*, and *Atrina rigida* (Supplementary Figs. 10–14). The mRNA expression of the four SMPs encoding genes were all significantly upregulated under the 1.5% $CO_2$ exposure compared to the control groups. The increased mRNA expression levels were 8.70± 5.01-fold ($p = 0.003307$), 9.85± 7.64-fold ($p = 0.0228$), 8.31± 3.93-fold ($p = 0.04893$), and 7.98± 2.243-fold ($p = 0.04157$) for *Cv-Nacrein*, *Cv-Pif97*, *Cv-Chits*, and *Cv-Tyr*, respectively (Fig. 2E–H).

**Elevated *p*$CO_2$ disturbed early shell formation and shell organic matrix biosynthesis in vivo**

To investigate the effects of elevated *p*$CO_2$ on early shell formation in vivo, developmental morphology and shell composition of *C. virginica* larvae were examined across critical early stages. A 1000 ppm atmospheric $CO_2$ environment was established within a sealed glovebox system to mimic the OA condition during larval shell development (Supplementary Fig. 1A, B). After 48 h of $CO_2$ exposure, the average seawater pH decreased significantly from 7.85 under the control condition (425 ppm $CO_2$) to 7.30 under the 1000 ppm $CO_2$ treatment, indicating the effectiveness of the OA simulation system (Supplementary Fig. 1D). With increased *p*$CO_2$ exposure, the early shell development during the period from trochophore (12 h old) to D-shaped (36 h old) *C. virginica* larvae (Supplementary Fig. 1C) exhibited clear morphological differences. The trochophore larvae failed to develop into a D-shape morphology and showed asymmetric shell shape (Fig. 3A, B). Larval shell deformity patterns, including concave hinge and protruding mantle on the fringe[75,76] were also detected under elevated *p*$CO_2$ concentration (Supplementary Fig. 4). In addition, the combination of Calcofluor/Calcein staining was utilized to assess the production of organic matrix and $CaCO_3$, respectively. Compared to the distribution of the calcofluor signal (blue, indicator of organic matrix) on the outer shell surface of the

larvae in control group, the location of calcofluor staining in OA-treated D-shaped larvae dispersedly distributed across the entire shell surface (Fig. 3A, B). Notably, shell calcification also irregularly displayed a brighter calcein (green, indicator of calcium) signal around hinge areas under simulated OA conditions.

**Abnormal gene expression profiles of early shell formation under elevated atmospheric *p*$CO_2$ during *C. virginica* larval development**

At the four primary developmental stages of *C. virginica* larvae (Supplementary Fig. 1C), including trochophore (T, 12 h), D-shape (D, 36 h), umbonal (U, 10 days), and pediveliger (P, 20 days) stages, the transcriptional pattern of *Cv-CaM*, *Cv-CaN*, and four SMPs biomarkers encoded genes showed differential responses to the elevated $CO_2$ environment (Fig. 3C, D). Although the upregulation of *Cv-CaM* by elevated *p*$CO_2$ condition was observed at the trochophore (3.01± 1.42 fold of control, $p = 0.028$) and the pediveliger stages (2.40± 1.64 fold of control, $p = 0.048$), there was downregulation of the CaM expression pattern at the D-shape stages (0.84± 0.22 fold of control, $p = 0.033$) and no significant changes in the umbonal stages (Fig. 3A). On the other hand, significant downregulation of *Cv-CaN* was observed in all the larval development stages under the OA simulation (fold changes ranging from 0.47± 0.27 to 0.86± 0.32, $p < 0.05$), except D-shape larvae, which showed similar expression pattern between control and OA treatment groups (Fig. 3D). While the mRNA levels of *Cv-Nacrein* and *Cv-Pif97* showed no significant changes at the D-shape and the pediveliger stages, they were both significantly upregulated at the trochophore stage (Nacrein: 3.76± 2.13 fold of control, $p = 0.016$; Pif97: 3.56± 1.64 fold of control, $p = 0.041$) and downregulated at the umbonal stage (Nacrein: 0.36± 0.20 fold of control, $p = 0.008$; Pif97: 0.33± 0.17 fold of control, $p = 0.017$) under the OA simulation (Fig. 3E, F). Compared with the control group,

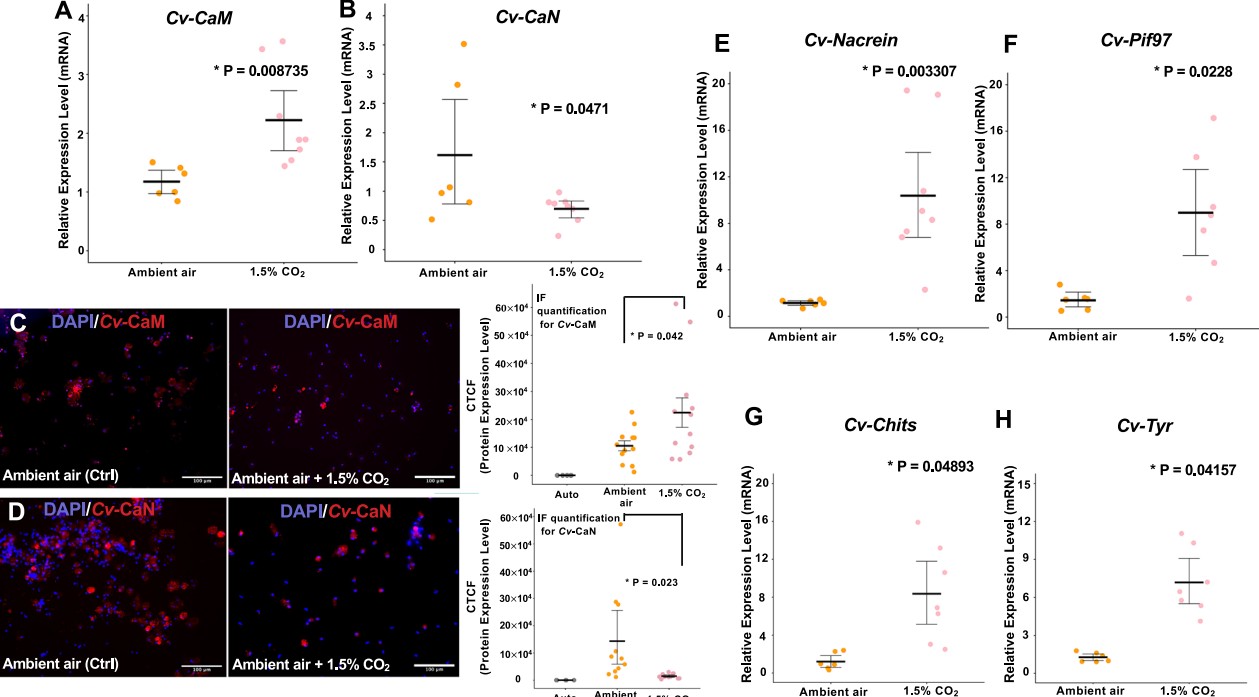

**Fig. 2 | Responses of calcium-signaling-related genes and SMPs in *C. virginica* mantle cells under simulated CO₂ conditions.** The relative mRNA expression of *Cv-CaM* (**A**) and *Cv-CaN* (**B**) in *C. virginica* mantle cells under simulated CO₂ conditions. Protein levels of *Cv-CaM* (**C**) and *Cv-CaN* (**D**) were compared among autofluorescence (Auto), control, and 1.5% CO₂ exposure with IF quantification via ImageJ (*n* = 3 images per oyster). The relative mRNA expression level of *Cv-Nacrein* (**E**), *Cv-Pif97* (**F**), *Cv-Chits* (**G**), and *Cv-Tyr* (**H**) in *C. virginica* mantle cells between ambient air and elevated pCO₂ conditions (*n* = 6–8 oysters). All CO₂ treatments were applied to cells for 48 h. *P < 0.05, a Student's *t* test was used for two-group comparisons, while a one-way ANOVA followed by Westfall's post hoc test was used for multiple comparisons. White scale bar: 100 μm. Error bar: 95% confidence interval of the mean.

expression of *Cv-Tyr* in the CO₂-treated larvae only displayed significant downregulation at the umbonal stage (0.43± 0.22-fold of control, $p = 0.004$, Fig. 3G). Chitin synthase, another SMP regulating the synthesis of extracellular matrix components, showed significant downregulation at the trochophore (0.73± 0.24-fold of control, $p = 0.049$) and umbonal stages (0.47± 0.25-fold of control, $p = 0.0007$) for the OA treatment but no significant changes at the D-shape and pediveliger stages (Fig. 3H).

**Elevated pCO₂ treatment and pharmacological calcium-CaM signaling disruption led to comparable dysregulation of calcium-signaling-related genes and SMPs-encoding genes in vivo**

The results of elevated pCO₂ treatment above showed that the simulated OA condition not only dysregulated the calcium-signaling pathway but also affected shell development by changing the expression of SMPs. Thus, we assumed the modified expression of SMPs resulted from the dysregulated calcium-signaling pathway under increased pCO₂ concentration and ultimately led to shell deformation. To test this hypothesis, we assayed the relationship between the calcium-signaling pathway and SMPs expression in the mantle epithelial cells by inhibiting the calcium-CaM binding site using W-7. Similar to the mantle cells under elevated pCO₂ exposure, the cells treated by W-7 also demonstrated significant upregulation of CaM and downregulation of CaN at both mRNA and protein levels (Fig. 4A, B, G, H). With 25 μM W-7 treatment, the mRNAs of *Cv-CaM* were upregulated by 4.10± 2.31-fold ($p = 0.04348$, Fig. 4A). Compared to the control group, there was 0.4897± 0.365-fold ($p = 0.04257$) downregulation of *Cv-CaN* under the W-7 treatment at the mRNA level (Fig. 4B). Additionally, the significant upregulations of *Cv-Nacrein* (13.75± 10.074-fold of control, $p = 0.037$), *Cv-Pif97* (12.71± 11.051 fold of control, $p = 0.0417$), *Cv-Tyr* (17.26± 8.309-fold of control, $p = 0.0043$) and *Cv-Chits* (5.70± 3.489 fold of control, $p = 0.041$) were also detected in the mantle cells under the W-7 treatment (Fig. 4C–F), which is consistent with their mRNA expression pattern under the OA simulation treatment.

Similarly, there was 112.5± 29% higher protein production of CaM in the *C. virginica* mantle cells under the W-7 treatment than in the control group (Fig. 4G). In contrast to the CaM protein level, the protein level of CaN under the W-7 treatment decreased by 94.61 ±58% compared with the cells under the control condition (Fig. 4H). Notably, with the blockage of CaM binding site on calcium by W-7, the phosphatase enzymatic activity of CaN was significantly decreased (0.511± 0.097-fold of control, $p = 0.00208$, Fig. 4I). Likewise, the elevated CO₂ level also reduced the phosphatase enzymatic activity of CaN to 0.534± 0.022-fold of control in the mantle cells ($p = 0.00151$, Fig. 4I).

**Inhibition of calcium-CaM signaling disrupted the early shell formation of D-shape *C. virginica* larvae**

With the W-7 blockage of calcium-CaM binding, *C. virginica* larvae also demonstrated disorganization of shell organic matrix production, similar to those larvae in the elevated pCO₂ experiment. After 24 h of W-7 treatment, the organization of the organic shell matrix, indicated by the calcofluor signal (blue), exhibited a distinct pattern in the D-shape larvae (Fig. 5A–D). In the control group, the calcofluor signal was visibly concentrated in the center of the shell field and gradually faded toward the front end of the shell edge (Fig. 5A). Under 1 μM W-7 treatment, the calcofluor signal was more uniformly distributed across the whole shell surface and showed higher intensity in the center (Fig. 5B). In contrast, larvae exposed to 2 μM W-7 displayed a reduced calcofluor signal compared to the above treatments. The organic matrix appeared to have irregular patterns, with asymmetric growth of the organic matrix and lower intensity of the calcofluor signal compared to the control group. However, there was no obvious malformation of the calcification process in the oyster shell (Fig. 5C). A dramatic change was detected under 5 μM W-7 treatment, where the shell exhibited substantial calcein signal around the center. Nevertheless, the organic matrix was confined to a thin layer along the margins of the valve, with a faint calcofluor signal observed at the center (Fig. 5D).

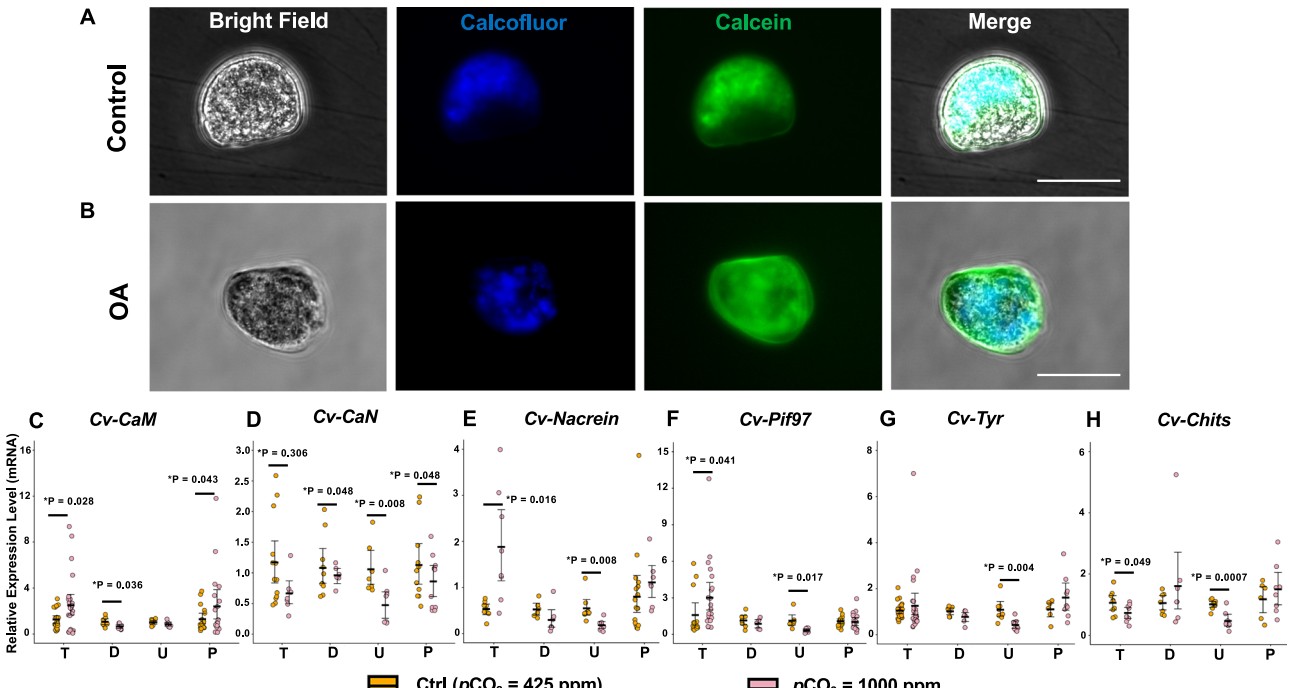

**Fig. 3 | Morphological and transcriptional responses of *C. virginica* larva early shell formation to the simulated OA. A** Morphological characteristics, organic matrix production (calcofluor staining), and $CaCO_3$ deposition (calcein staining) of early D-shape larvae (36-hour-old) under control conditions. **B** Morphological characteristics and staining of early D-shape larvae under OA conditions. **C** mRNA expression pattern of the calcium-signaling-related gene *Cv-CaM*. **D** mRNA expression pattern of the calcium-signaling-related gene *Cv-CaN*. **E** mRNA expression pattern of the SMP gene *Cv-Nacrein*. **F** mRNA expression pattern of the SMP gene *Cv-Pif97*. **G** mRNA expression pattern of the SMP gene *Cv-Tyr*. **H** mRNA expression pattern of the SMP gene *Cv-Chits*. Gene expression was measured across developmental stages: T (trochophore), D (D-shaped), U (umbonal), P (pediveliger) ($n = 8$–21 1-mL aliquots). Trochophore stage larvae were exposed to elevated $CO_2$ for 24 h, and the rest of the larval stages were exposed for 48 h. *$P < 0.05$, one-way ANOVA followed by Westfall's method for multiple comparison. White scale bar: 100 μm. Error bar: 95% confidence interval of the mean.

The ratio between matrix-occupied and calcified areas in a single valve from each larva was measured during early larval development. Despite there being no significant difference in the average ratio of matrix to calcified areas between the larvae in the control group and 1 μM W-7 treatment groups, the ratio significantly reduced by 57.22% under 2 μM and 45.31% under 5 μM W-7 treatment compared to the control group (Fig. 5E). Notably, compared with the calcofluor signal intensity on the shell surface in the control group, the average fluorescent intensity was significantly enhanced by 59.79% under 1 μM W-7 treatment. In contrast, the average intensity decreased substantially by 56.34% and 48.80% in the larvae exposed to 2 μM and 5 μM W-7 treatments, respectively (Fig. 5F).

### Rescue of calcium-signaling pathway and SMP production in mantle cells under CO₂ stress by CaN

In the *C.virginica* mantle epithelial cells, both elevated $pCO_2$ and W-7 experiments displayed decreased protein production and phosphatase activity of CaN due to the dysregulated calcium-signaling pathway. Based on this result, we further determined that the inhibition of CaN under the elevated $pCO_2$ concentration affected the SMPs expression using a CaN addition rescue experiment. Expression of the *Cv-CaM* and four SMPs in $CO_2$-exposed *C. virginica* mantle cells in the presence of CaN were consistent with the mantle cells under ambient air conditions. The mRNA expression of *Cv-CaM* in the $CO_2$-treated mantle cells with 20U ($p = 0.0001$) and 100U ($p = 0.0025$) CaN addition was significantly downregulated to control levels compared to the cells without CaN treatment, respectively (Fig. 6A). However, *Cv-CaN* mRNA expression showed no significant difference between cells treated with elevated $CO_2$ alone and those exposed to both $CO_2$ and CaN (Fig. 6B). Compared to mantle cells exposed to increased $CO_2$ without CaN treatment, *Cv-Nacrein* mRNA expression in $CO_2$-exposed mantle cells was significantly downregulated with 20U

($p = 0.00153$) and 100U ($p = 0.00164$) CaN addition (Fig. 6C). Similarly, *Cv-Pif97* mRNA expression in $CO_2$-exposed mantle cells treated with 20U and 100U CaN returned to control levels, significantly lower than in the mantle cells treated with elevated $CO_2$ alone (20U CaN: $p = 0.00579$; 100U CaN: $p = 0.01577$) (Fig. 6D). Under elevated $CO_2$ conditions, *Cv-Tyr* mRNA expression in mantle cells with 20U ($p = 0.0495$) and 100U ($p = 0.0414$) CaN addition was significantly lower than in mantle cells without CaN treatment (Fig. 6E). Furthermore, the *Cv-Chits* also displayed a significantly reduced pattern at the mRNA level in $CO_2$-stimulated mantle cells treated with CaN compared to those exposed to $CO_2$ alone (20U CaN: $p = 0.0441$, 100U CaN: $p = 0.0485$) (Fig. 6F). There was also no significant difference between the $CO_2$-treated cells in the presence of CaN and the mantle cells in the control group.

### Discussion

Despite calcium being the most abundant mineral element in the marine bivalve shell structure, its role extends beyond providing substrate resources for biomineralization, as it also regulates various physiological processes as a secondary messenger intracellularly[77–79]. Under the stress of elevated $pCO_2$, the dysregulation patterns of intracellular calcium concentration are observed in oyster hemocytes[60,61], suggesting that marine bivalves experience fluctuations in both intracellular and extracellular calcium levels in response to changes in carbonate chemistry in the ocean induced by the OA conditions. Recent studies highlight the crucial role of the calcium-CaM signaling pathway in vertebrate bone formation by regulating osteoblast differentiation and proliferation[39,41,80,81]. Several important calcium-signaling-related genes, including CaM and CaN, are reported to be involved in shell formation in marine bivalves[17,18,35,38]. More interestingly, the amino acid sequences of *C. virginica* CaM and CaN exhibited high similarity to those of CaM and CaN from bivalves, gastropods, arthropods,

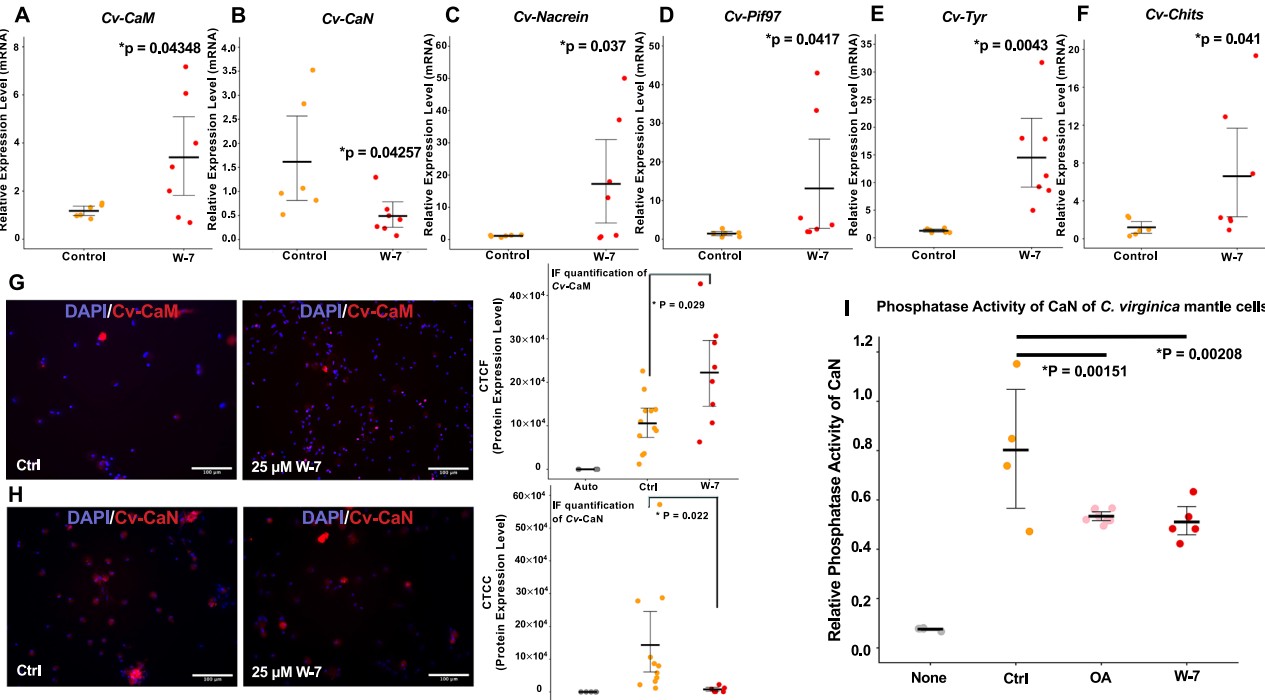

**Fig. 4 | Impact of CaM inhibitor (W-7) and OA condition on calcium signaling and biomineralization-related markers gene expression and CaN phosphatase activity in *C. virginica* mantle epithelial cells.** (A–F) Expression of *Cv-CaM* (**A**), *Cv-CaN* (**B**), *Cv-Nacrein* (**C**), *Cv-Pif97* (**D**), *Cv-Tyr* (**E**), and *Cv-Chits* (**F**) in *C. virginica* mantle cells between ambient air and W-7 treatment (n = 6–8 oysters). (G-H) Protein levels of Cv-CaM (**G**) and Cv-CaN (**H**) were respectively compared among autofluorescence (Auto), control, and 1.5% $CO_2$ exposure with IF quantification via ImageJ (n = 3 images per oyster). (**I**) CaN phosphatase activity under the simulated OA condition (1.5% $CO_2$) and the W-7 treatment (25 μM) (n = 4–6 oysters). All W-7 and $CO_2$ treatments were applied to cells for 48 h. *$P < 0.05$, a Student's *t* test was used for two-group comparisons, while a one-way ANOVA followed by Westfall's post hoc test was used for multiple comparisons. White scale bar: 100 μm. Error bar: 95% confidence interval of the mean.

and vertebrates (Supplementary Figs. 8, 9). These alignment analyses suggested that the calcium/CaM/CaN signaling play a conserved role in regulating the biomineralization process across taxa. In this study, we identified an intracellular calcium influx pattern in oyster mantle epithelial cells, triggering dysregulation of the calcium-CaM signaling pathway under elevated $pCO_2$ conditions. As a result of the calcium-CaM signaling pathway disruption, dramatic morphological changes in shell shape and the shell organic matrix were observed, which are consistent with previous reports of OA effects on shell formation[75,76].

To link calcium signaling with the genetic regulation of *C. virginica* shell formation in response to OA, we established a more efficient in vitro model using oyster mantle cells. Compared to previous methods for isolating mantle cells from oysters using cell dissociation and earlier mantle tissue explant techniques[34,82], our method pronouncedly reduced the populations of other cell types, such as fibroblast cells and hemocytes (Supplementary Fig. 3). The high-quality, long-term in vitro maintenance of oyster mantle epithelial cells in this study provided a more robust system for monitoring prolonged, low-level environmental changes.

Our group's previous study showed that four calcium-binding proteins, including Cv-CaM, exhibited dose-dependent transcription in oyster mantle cells under different $CO_2$ exposure conditions (1% and 2.5%)[34]. Therefore, based on these results, a 1.5% $CO_2$ level was selected to explore the potential molecular mechanisms and signaling pathways regulating shell formation under elevated $CO_2$. To maintain stable pH conditions for primary cell culture experiments, HEPES and NaHCO3 (Supplementary Table 1) were added to the oyster cell culture medium, resulting in a pH drop from 7.8 to 7.4 within 48 h of treatment (Supplementary Fig. 6). It suggested that, compared to using environmentally relevant $pCO_2$ concentrations (e.g., 0.1% $CO_2$) for in vitro studies, an extremely high $pCO_2$ exposure is a more appropriate level to induce measurable cellular and molecular responses in mantle cells. Besides, the calculated carbonate chemistry

parameters ([$HCO_3^-$] and [$CO_3^{2-}$]) in the cell culture medium under elevated $pCO_2$, were surprisingly similar to those reported for bivalve EPF, under acidified seawater (Supplementary Table 4)[55]. This suggests that the high $pCO_2$ level applied in the in vitro study may reflect actual cellular exposure conditions, as the resulting pH and carbonate chemistry are within the physiological range relevant to OA studies.

Under elevated $pCO_2$ conditions, a corresponding increase in intracellular calcium concentration in the mantle epithelial cells (Fig. 1E–H) is consistent with an increased trend of extracellular calcium in the hemolymph of adult oysters exposed to elevated $pCO_2$ for 60 days[61]. These results suggest that the elevated $CO_2$ level may induce calcium influx in the oyster mantle epithelial cells as a compensatory response to maintain calcium homeostasis in fluctuating ambient conditions[63]. Along with CaM, several calcium-binding proteins have been shown to be upregulated in the oyster mantle under elevated $CO_2$ conditions[34,61,83]. Several studies indicate that the OA conditions disrupt the carbonate system and calcium level in marine bivalves[51,55]. Consequently, this study suggested that the dynamic of calcium concentrations in the intracellular and extracellular environments of oyster mantle epithelial cells triggered the differential expression of calcium-binding proteins and alteration of the calcium-signaling pathway.

In the calcium-signaling pathways, CaM plays an essential role in various cellular processes in marine bivalves, such as the processes of environmental stress acclimation[84], immune response[85], and neuroendocrine regulation[86]. However, most studies on the biomineralization-related function of CaM focus on its role as an SMP in organizing the structure of shell layers and the transformation of $CaCO_3$ crystals during marine bivalve shell development. Our study demonstrated the upregulation of CaM at both mRNA and protein levels in isolated mantle epithelial cells under increased $pCO_2$ exposure (Fig. 2A–D). These results are consistent with previous findings that reported upregulation of several CaM isoforms in oyster mantle tissue following $CO_2$ exposure in adult oysters[83] and at the

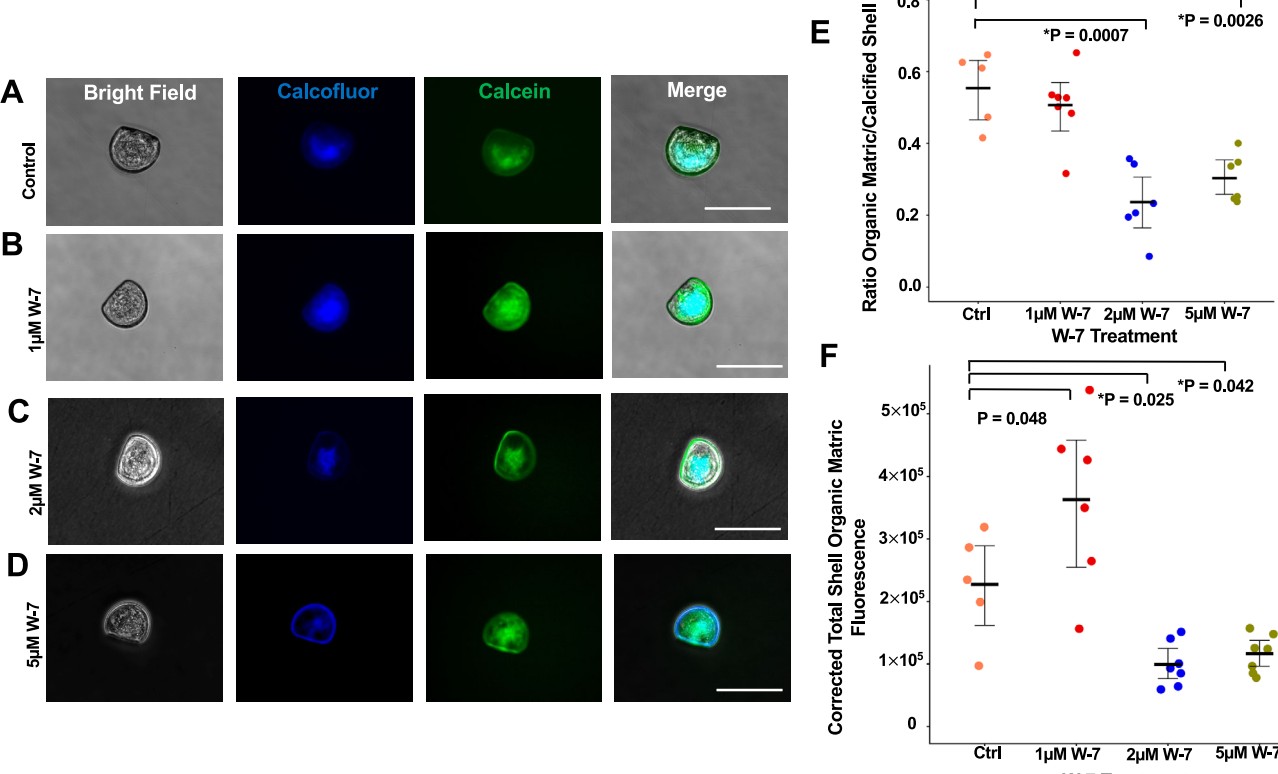

**Fig. 5 | Effect of CaM inhibitor (W-7) on early shell development from trochophore to D-shape stages in *C. virginica*. A** Brightfield, calcofluor fluorescent staining (Shell organic matrix), calcein fluorescent staining (CaCO₃ deposition), and merged images of early D-shaped oyster larvae under control conditions. **B** Imaging of larvae under 1 μM W-7 treatment. **C** Imaging of larvae under 2 μM W-7 treatment. **D** Imaging of larvae under 5 μM W-7 treatment. **E** Ratio of areas occupied by the shell matrix and calcified shell in the *C. virginica* D-shaped larvae following different W-7 treatments. **F** Fluorescent intensity of calcofluor corresponding to shell organic matrix production in *C. virginica* D-shaped larvae after different W-7 treatments ($n = 5–7$ larvae). All the larvae were treated with W-7 for 24 h. *$P < 0.05$, one-way ANOVA followed by Westfall's method for multiple comparisons. White scale bar: 100 μm. Error bar: 95% confidence interval of the mean.

cellular level[34]. However, in our in vivo oyster larvae models, only the trochophore and pediveliger stages exhibited upregulation of CaM in response to OA conditions (Fig. 3C). Notably, these two stages are involved in initial shell formation and the preparation of metamorphosis for settlement[87]. Further, several CaM-like proteins have shown downregulation patterns in mantle tissue due to a potential calcium compensation strategy among the gill and mantle induced by CO₂ exposure[63]. Given the diverse cellular functions of CaM at the organism level, the in vitro mantle epithelial cell model demonstrated a higher efficiency and specificity in revealing the responses of the mantle epithelium cellular processes to the stress of simulated acidification conditions.

As an essential downstream effector of the calcium-CaM signaling pathway, CaN is revealed to play an important role in marine bivalve shell formation by regulating the BMP signaling pathway[38]. In a previous study, two CaN subunits were isolated from the pearl oyster *P. fucata*: subunit A, which functions as the phosphatase catalysis unit, and subunit B, which serves as the calcium-binding site[30]. To explore the role of phosphatase activity in CaN during shell formation, we used the CaN subunit A homolog from *C. virginica* to assess the responses of phosphatase-related genes to OA treatment (Supplementary Table 3). CaN expression was downregulated at the mRNA and protein levels in oyster mantle cells with increased CO₂ exposure (Fig. 2B, D). Similarly, CaN downregulation was observed across oyster larval development stages in acidified conditions (Fig. 3D), which is consistent with a previous study documenting the decreased transcription level of CaN in *M. gigas* D-shaped larvae exposed to decreased pH stress[66]. These shared patterns among the in vitro and in vivo models suggest that the dysregulation of the calcium-CaM signaling pathway is induced by elevated $p$CO₂.

In addition to the dysregulation of calcium-CaM, $p$CO₂ exposure also affected the expression of four conserved SMPs associated with bivalve shell formation in both in vitro and in vivo oyster models (Figs. 2 and 3). The upregulation of *Cv-Nacrein* in oyster mantle epithelial cells suggests a potential reduction of oyster calcification rate[11,71]. While in vivo analysis also revealed an upregulation pattern of *Cv-Nacrein* during initial shell formation at the trochophore stage (Fig. 3E), these findings contrasted with previous studies that significant nacrein upregulation was only detected after the pediveliger stage[72,88]. The unexpected early upregulation was potentially related to a disruption of crystal formation during shell construction[71,89]. Since Pif97 regulates shell formation by inhibiting calcite crystal growth and amorphous CaCO₃ stabilization[13,90], the increased expression level of this gene in the mantle epithelial cells suggested an OA-induced disruption of shell framework construction. The consistent upregulation of *Cv-Pif97* during trochophore development under $p$CO₂ exposure indicated a negative impact of OA on the transformation and crystallization of CaCO₃ for early shell formation[74,91,92].

Compared to the upregulation of chitin synthase in the *C. virginica* mantle epithelial cells under OA conditions (Fig. 2G), the *Cv-Chits* at both trochophore and umbonal stages exposed to elevated $p$CO₂ exhibited the opposite expression pattern (Fig. 3H). Previous studies find chitin decomposition during early shell formation in acidified-treated trochophore oysters[74,93]. Given the evolutionary diversity of chitin synthase isoforms among mollusks, the contrasting *Cv-Chits* expression between the in vitro and in vivo models under OA suggests alternative biological functions of chitin synthetase beyond the biomineralization process in adult *C. virginica*[94]. Similarly, the tyrosinase (*Cv-Tyr*) also showed contrasting expression patterns between in vivo and in vitro models. While there was no

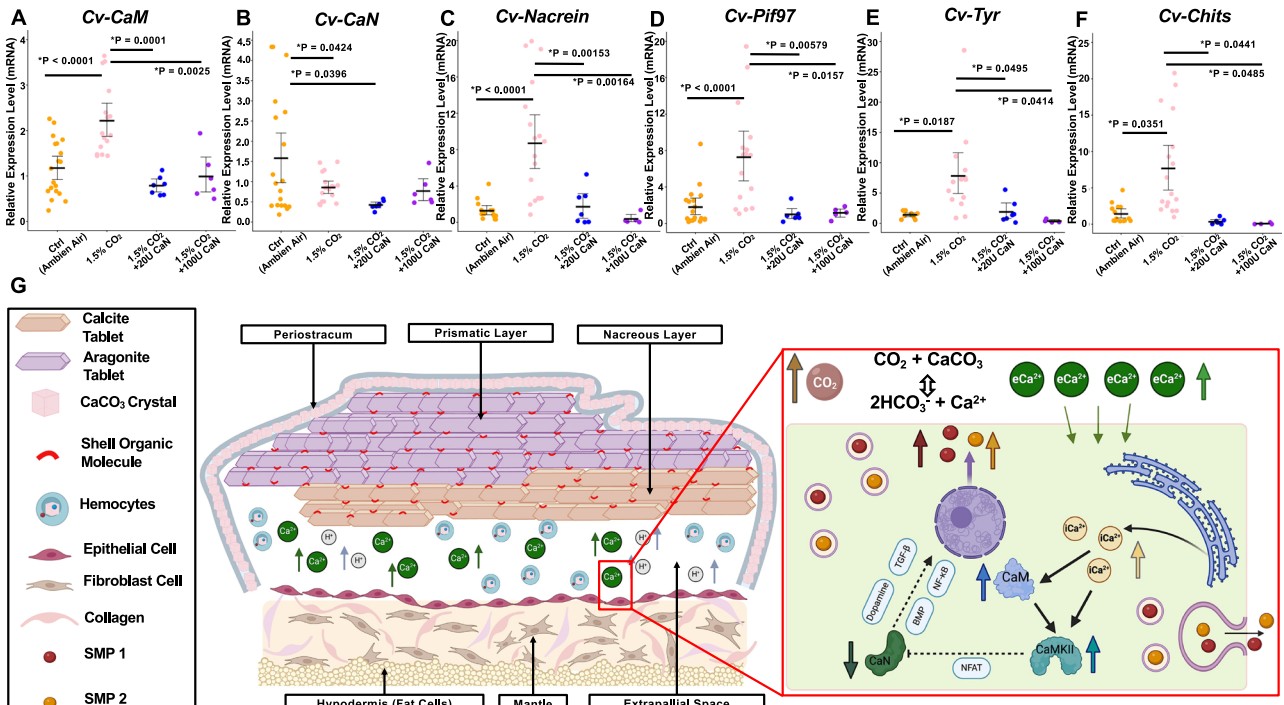

**Fig. 6 | Impact of CaN on SMP expressions and a mechanistic model of calcium-CaM dysregulation in *C. virginica* mantle cells under elevated $pCO_2$.** Functional analysis of different CaN concentrations (20U and 100U) in regulating the transcription of SMPs in *C. virginica* mantle epithelium cells under the elevated $pCO_2$ condition. Comparison of relative mRNA expression of *Cv-CaM* (**A**), *Cv-CaN* (**B**), *Cv-Nacrein* (**C**), *Cv-Pif97* (**D**), *Cv-Tyr* (**E**), and *Cv-Chits* (**F**) in the elevated $CO_2$ exposed *C. virginica* mantle cells with and without the presence of additional CaN ($n = 4–8$ oysters). Significant difference between the two groups ($P < 0.05$) was demonstrated by different letters over the column. All $CO_2$ treatments with or without the addition of CaN were applied to cells for 48 h. *$P < 0.05$, one-way ANOVA followed by Westfall's method for multiple comparisons. Error bar: 95% confidence interval of the mean. **G** Schematic model illustrating the potential impact of OA on the dysregulation of the calcium-CaM signaling pathway and disrupting the organization of SMPs in *C. virginica* mantle epithelium cells (Created in BioRender, https://BioRender.com/twqli3l and https://BioRender.com/ab7h0b4).

significant impact of elevated $CO_2$ stress on the expression of *Cv-Tyr* at the trochophore stage (Fig. 3G), the upregulation of *Cv-Tyr* in the mantle cells under the OA stress (Fig. 2H) is in accordance with the increased mRNA expression of tyrosinase at the trochophore stage in *Crassostrea angulata* responding to OA[95]. The upregulation of *Cv-Tyr* revealed a potential disruption of the shell matrix production of oysters[70]. The reversed expression pattern of *Cv-Tyr* in larval *C. virginica* at the umbonal stage indicated a potential deformation of periostracum, the initial shell layer during the oyster's early development[96,97], which suggests potentially different shell formation mechanisms compared to adult oysters. Collectively, these findings indicate that elevated $pCO_2$ disrupts multiple aspects of shell biomineralization. The variable transcriptional level of the SMPs during *C. virginica* larval development also implied potentially different acclimation strategies for the shell formation process across larval stages due to the different forms of mineralization related to the developmental phases of oyster larvae[72,98].

In the in vivo study, neither D-shape larvae nor veliger juveniles showed any significant changes in shell sizes under the mimic OA system within the glovebox and 1000 ppm atmosphere $CO_2$ (Supplementary Fig. 5), contrasting with previous studies that stimulate OA conditions by directly bubbling $CO_2$ into seawater[74,95,99]. Compared to previous methods, the OA simulation system developed in this study more realistically mimicked the actual air–sea $CO_2$ exchange in nature. Direct $CO_2$ delivery to seawater may cause rapid and uneven changes in seawater chemistry, leading to artificially low pH levels and excessive shell dissolution that may not accurately represent natural ocean acidification processes. Despite the lack of prominent shell size changes, disarrangement of shell organic matrix and larval shell deformity patterns were detected (Fig. 3A, B and Supplementary Fig. S4). Moreover, reductions in $[CO_3^{2-}]$, aragonite ($\Omega Ar$), and calcite ($\Omega Ca$) saturation states displayed similar trends observed in conventional $CO_2$-

bubbling acidification systems (Supplementary Table 4)[100]. These findings demonstrated the effectiveness of our OA simulation system in replicating future acidified marine conditions without introducing over-acidified environments. Furthermore, seawater exposed to increased atmospheric $CO_2$ in the sealed glovebox induced a significant decrease in pH after two days of treatment (Supplementary Fig. 1D).

This study also illustrated a potential connection between the dysregulated calcium-CaM signaling pathway and the abnormal expression of biomineralization-related SMPs in oysters under OA conditions. By blocking the calcium-binding sites on CaM, the CaM antagonist W-7 induced mantle epithelial cells to produce CaM while suppressing CaN protein production. The W-7 exposure functioned similarly to OA stress (Fig. 2 and 4). In addition, decreased phosphatase activity of CaN in the *C. virginica* mantle epithelial cells was also detected under both the elevated $pCO_2$ and the W-7 treatments (Fig. 4I). These results suggested that dysregulation of the calcium-CaM signaling pathway inhibited CaN activity. Although no direct evidence linked CaM to the expression of CaN, a possible explanation involves the upregulation of calcium/CaM-dependent protein kinase II (CaMKII), which was detected in both $CO_2$-exposed and W-7-treated *C. virginica* mantle cells (Supplementary Fig. 7). CaMKII activation is found to inhibit the production of the Nuclear Factor of Activated T-cells (NFAT), which is identified as a transcription factor of CaN expression[101]. Interestingly, the activity of NFAT is also triggered by the dephosphorylation through CaN[102]. Besides, the in vivo analysis of the W-7 treatment on early shell development in D-shape oyster larvae also demonstrated significant disorganization of the shell organic matrices (Fig. 5) and the shell growth (Supplementary Fig. 5). It further illustrated the role of the calcium/CaM signaling pathway in driving oyster shell construction via regulating SMPs production and arrangement.

As the hypothesized downstream products of the calcium-signaling pathway, the four SMPs in mantle epithelial cells were also regulated consistently between the treatments of the W-7 and increased $p\mathrm{CO_2}$. The anomalous changes in the organic matrix were attributed to the abnormal production of SMPs, which organize the further calcification step by transforming and organizing $\mathrm{CaCO_3}$ deposition[103]. The presence of the chitin-binding domains in the Pif97[13] and tyrosinase[104] and a calcium-binding domain in nacrein[71] further suggests the role for these SMPs in oyster shell construction. Collectively, these results highlight the negative impact of the dysregulated calcium-CaM pathway on the production of SMPs and the organization of the organic matrix framework on the oyster shell surface.

The CaN rescue experiments demonstrated that elevated CaN levels in the oyster mantle cells resulted in upregulated SMP expression in the $\mathrm{CO_2}$-treated cells returning to the control levels (Fig. 6C–F). The reduced CaN activity under OA conditions was linked in this study to the transcription of SMP encoding genes, including *Cv-Nacrein*, *Cv-Pif97*, *Cv-Tyr*, and *Cv-Chits* in *C. virginica*. As mentioned above, the decreased CaN production was possibly caused by the suppression of NFAT activity[105,106], which interferes with several biomineralization-related signaling pathways, such as the BMP/Smads[38,107], Wnt[108], and NF-κB signaling pathways[109]. For example, previous studies show that Rel/ NF-κB mediates the transcription of nacrein in *P. fucata*[28,110]. The transcription of Pif97 is demonstrated to be related to the BMP/Smads signaling pathway[26,111]. The inhibition of CaN is found to decrease the biosynthesis of dopamine[112] and subsequently affect the synthesis of tyrosinase and chitin[93]. Surprisingly, the transcription levels of *Cv-CaN* in the $\mathrm{CO_2}$-treated mantle cells, following the addition of CaN, displayed a similar expression pattern to mantle cells without CaN treatment (Fig. 6B). One possible explanation for the reduced CaM expression in the $\mathrm{CO_2}$-exposed mantle cells treated with CaN (Fig. 6A) is that the external introduction of CaN was activated by binding to calcium-bound CaM. This binding likely decreased the availability of elevated free calcium, which in turn led to lower CaM gene expression.

Based on the results of this study, we propose a model of the calcium-CaM signaling pathway regulating SMP production to explain the impact of future OA conditions on the biomineralization process in *C. virginica* (Fig. 6G). Under OA stress, the dissolution of shell in marine bivalves leads to increased extracellular calcium levels and carbonate ions in the extrapallial fluid to support calcification[52,55,61]. To maintain the calcium homeostasis between the intra- and extracellular environment, the mantle epithelial cells of marine bivalves release large amounts of calcium intracellularly and enhance the calcium transportation from the ambient environment through the cell membrane[60,62,86]. To counteract shell dissolution under long-term OA stress, oysters are shown to increase extracellular calcium (serum level) and upregulate CaM production, which plays a critical role in calcium deposition and transportation[61,63]. However, at the same time, this persistent upregulation of CaM may dysregulate the calcium-CaM signaling pathway by inhibiting the activities of CaN-NFAT and triggering the CaMKII-CREB pathway due to elevated concentrations of the calcium-bound CaM complex. Similar effects are suggested in human cardiac myocytes[101]. Finally, the decreased CaN activity results in the abnormal expression pattern of several SMPs by interrupting relevant signaling pathways, including the BMP signaling pathway (Pif97[26,111]), NF-κB signaling pathway (nacrein[28]), and TGF-β signaling pathway (chitin synthase and tyrosinase[93]).

## Conclusion

In summary, our results identified increased intracellular calcium and the dysregulation of calcium-CaM in the *C. virginica* mantle epithelial cells under a simulated OA environment as a strategy to maintain calcium homeostasis. The persistent calcium regulation, driven by the upregulation of CaM, reduced CaN protein production and phosphatase activity in the oyster mantle epithelial cells. The reduced CaN activity induced by elevated $\mathrm{CO_2}$ exposure disrupted the production of various SMPs involved in the shell formation process and, consequently, disorganized the shell organic

matrix framework and the shell construction process. This study clarifies the role of the calcium-signaling pathway in marine bivalve shell formation. These findings, especially the essential role of CaN in regulating oyster SMP production and shell organic matrix organization, serve as valuable data for understanding the cellular response to OA conditions induced by elevated $p\mathrm{CO_2}$ in marine bivalves and for developing potential strategies for mitigation.

## Methods

### Oyster mantle cell culture

Adult *C. virginica* oysters of south Texas origin were obtained from the Texas A&M AgriLife Research Mariculture Facility (Corpus Christi, TX) and acclimated for two weeks in a recirculating system with 1 μm-filtered seawater from Corpus Christi Bay, Texas (~30ppt salinity) at 22–24 °C. Before primary cell culture, oysters were incubated overnight in sterilized calcium and magnesium-free artificial seawater solution (CMFSS) (Supplementary Table 1)[34], supplemented with 100 units/ml of penicillin and 100 μg/ml of streptomycin (CMFSS-Pen/Strep, pH = 7.7).

Pre-treated oysters were dissected, and connective tissues along the mantle pallial were collected (Fig. 1A). Mantle tissues were rinsed with CMFSS-Pen/Strep, then filtered with a 100 μm cell strainer. The rinsed mantle pieces were digested with 0.2% collagenase Type I (Thermo Fisher Scientific) in oyster cell culture basal medium (Supplementary Table 1) to separate fibroblast cells from mantle explants. After 3 h of digestion with collagenase I at 37 °C, the tissues were collected and minced into 1–2 mm³ explants using a sterilized scalpel. Four to five explants were placed in a 25 mm³ tissue culture flask (Thermo Fisher Scientific) for 45 min to adhere. The explants were then covered with 1 mL oyster cell culture medium supplemented with 10% Fetal Bovine Serum (FBS, Avantor, US), 100 units/ml of penicillin, and 100 μg/mL of streptomycin (oyster cell culture working medium). Primary cultures were incubated at 28 °C in a humidified incubator without $\mathrm{CO_2}$ for 48 h to allow cell migration.

### Stimulation of oyster mantle cells with increased environmental $\mathrm{CO_2}$

*C. virginica* mantle cells were observed under a microscope on the 2nd day post-explantation (dpe) to confirm attachment. On the 3rd dpe, the medium was replaced with 1.5 ml fresh oyster cell culture medium ++ before the elevated $p\mathrm{CO_2}$ treatment. Cells from explants were transferred to an incubator filled with ambient air (Control) or 1.5% $\mathrm{CO_2}$ on the 5th dpe. A $\mathrm{CO_2}$-controlled incubator (VWR Water-Jacked $\mathrm{CO_2}$ Incubator–Model 2325) equipped with an automatic gas regulator and a $\mathrm{CO_2}$ sensor was utilized to maintain a 1.5% $\mathrm{CO_2}$ atmospheric environment. After 48-h exposure, mantle cells were harvested for RNA isolation using TRIzol Reagent (Molecular Research Center, Cincinnati) or fixed in 4% paraformaldehyde (PFA) (Thermo Fisher Scientific) for immunofluorescent staining. The explants were removed by filtering through 40 μm cell strainers before RNA isolation in TRIzol. Mantle cells from eight *C. virginica* oysters were used for the control and the $\mathrm{CO_2}$ exposure groups.

### Intracellular calcium level measurement in oyster mantle cells under elevated $p\mathrm{CO_2}$ treatment

Short-term (1 h) and long-term (24 h) $\mathrm{CO_2}$ exposure experiments were conducted to evaluate intracellular calcium changes under elevated $p\mathrm{CO_2}$ in *C. virginica* mantle cells. Intracellular calcium levels were measured using Fluo-4 AM (Thermo Fisher Scientific) following the manufacturer's protocol. This cell membrane-permeable dye can be retained in the cytoplasm and exhibits a 100-fold increase in fluorescence when it binds $\mathrm{Ca^{2+}}$ after cleavage of the acetoxymethyl ester (AM) by intracellular esterases. Briefly, a 2:1 ratio of 1 μM Fluo-4 AM and 1 μM Polyethylenimine (PEI) mixture was added to the oyster cell culture working medium and incubated for 30 min before staining.

For short-term exposure, the calcium dye was incubated with isolated mantle cells for 1 h at 28 °C on the 5th dpe. After washing twice with Dulbecco's phosphate-buffered saline (DPBS, Corning Inc.) to remove

excess calcium dye, the cells were incubated in fresh culture medium and exposed to ambient air or 1.5% $CO_2$ for 1 h in a dark incubator. The fluorescent intensity in the stained cells was quantified before and after the $CO_2$ exposure under a fluorescent microscope (Olympus CKX53). At least five 10x magnified images derived from one oyster were captured for calcium measurement. The fluorescent signal of intracellular calcium was measured by calculating the mean gray value of the cells in each image via ImageJ[113].

For long-term $CO_2$ exposure, untreated and 24-h 1.5% $CO_2$-exposed mantle cells were incubated with Fluo-4 AM in oyster cell culture medium for 1 h following the staining protocol above. After rinsing off the dye with DPBS, the stained mantle cells were incubated in fresh culture medium under ambient air or 1.5% $CO_2$ for 24 h. The treated oyster mantle cells were then harvested with 0.05% trypsin (Corning Inc.), and explants were removed using 40 μm cell strainers. Trypsinized mantle cells were rinsed and resuspended with DPBS. The intracellular calcium fluorescence in each cell was subsequently quantified by flow cytometry (BD Accuri™ C6 Plus) with excitation wavelength at 495 nm and emission wavelength at 518 nm, analyzing 10,000 events. Although the calcium dye gradually leaked out of cells over time, this approach enabled a robust comparison between control and $CO_2$-treated cells, as cells with higher intracellular calcium levels retained stronger fluorescent signals. Mantle cells isolated from four *C. virginica* oysters were used in short- and long-term $CO_2$ exposure experiments.

### Assay of oyster shell formation process under W-7 treatment in mantle cells

N-(6-aminohexyl)-5-chloro-naphthalene sulfonamide hydrochloride (W-7), a specific CaM antagonist inhibiting calcium-CaM binding site on its target proteins CaN[114], was used to investigate the role calcium-CaM signaling pathway in *C. virginica* mantle cells. The mantle cells were treated with 25 μM W-7 (EMD Millipore Corp, $IC_{50}$ = 28 μM for calcium-CaM-dependent phosphodiesterase inhibition) on the 5th dpe for 24 h. A 100 mM stock solution of W-7 was prepared by dissolving the W-7 powder in dimethyl sulfoxide (DMSO) and stored at −20 °C. The working concentration of 25 μM W-7 was prepared by diluting the stock solution in oyster cell culture working medium. Treated mantle cells were harvested for RNA isolation or immunofluorescent staining. The mantle cells from eight *C. virginica* oysters were used in the W-7 treatment group.

### Immunofluorescence analysis of CaM and CaN in oyster mantle cells

Immunofluorescence (IF) was used to visualize and quantify the expression of CaM and CaN at the protein level of *C. virginica* mantle cells in the control and treatment groups. After the $CO_2$ exposure or W-7 treatment, the mantle cells were rinsed with phosphate-buffered saline (PBS) and fixed in 4% PFA for 1 h. The fixed samples were then blocked with 10% lamb serum in 1% Triton-PBS (PBST) for 2 h at room temperature. Primary antibodies of CaM and CaN (purchased from the Developmental Studies Hybridoma Bank at the University of Iowa, Iowa City, IA) diluted in PBST (1:100) were applied to cover the samples in each culture flask, respectively and incubated at 4 °C overnight. The primary antibodies used in this study contained mouse-anti-fungus CaM and mouse-anti-human CaN. After removing the primary antibodies and rinsing with PBS, the secondary antibody (goat anti-mouse) conjugated with Alexa flour TM 594 (red) dye (Thermo Fisher Scientific) was diluted in PBST (1: 400) and incubated with the samples for 2 h in a dark area at room temperature. A blue nuclear dye, 4′,6-diamidino-2-phenylindole (DAPI, Biotium), was used to fluorescently stain the nuclear DNA to localize the individual cells with 1:5000 dilution in PBST. The samples were monitored under the Olympus BX53 fluorescent microscope after IF staining. Three pictures were captured for each culture flask under 20× magnification. The exposure time of the image was adjusted to 50 ms under the UV channel and 250 ms under the Texas Red channel. The same exposure time was used on all images of each antibody. Pictures were also taken from secondary antibodies-stained samples without the primary antibodies as a negative control to avoid the disturbance of autofluorescence.

The total fluorescent signal of CaM and CaN in each cell was measured using ImageJ[113]. The protein expression level was calculated as the corrected total cell fluorescence (CTCF)[34,115], which was calculated as

$$Integrated\ Density-(Area\ of\ selected\ cell \times Mean\ fluorescence\ of\ background\ readings).$$

The calculated protein expression was rounded off to ensure consistent data presentation.

### CaN phosphatase activity assay in *C. virginica* mantle cells

Post-treatment *C. virginica* mantle cells were harvested in 1 mL of 0.05% trypsin (Corning Inc.), centrifuged to form a pellet, and lysed in 50 μL of calcineurin assay buffer (Enzo) for protein isolation. Calcineurin phosphatase activity was measured using the calcineurin phosphatase assay system (BML-AK904, Enzo) with 3 μg of protein from mantle cells, following the manufacturer's instructions. The activity was quantified spectrophotometrically by measuring the release of free phosphate from the calcineurin-specific RII phosphopeptide substrate using the BIOTEK Cytation 5 image reader (Agilent).

### Larval oysters production and simulated acidification treatments

South Texas *C. virginica* oyster larvae were collected from the Texas A&M AgriLife Mariculture Center, following the center's spawning and culture protocols. Four early life stages of oysters, including trochophore, D-shaped, umbonal, and pediveliger stages, were collected respectively in 12 h, 48 h, 10 days, and 20 days post-fertilization. A light microscope was used to ensure that >70% of the larvae had reached the particular development stages (Supplementary Fig. 1).

Oyster larvae were exposed to two different $pCO_2$ conditions, including current $pCO_2$ (425 ppm) and predicted $pCO_2$ (1000 ppm) levels based on IPCC[42]. Each experimental trial involved larvae placed in 1-liter plastic beakers with 700 mL of filtered seawater. A simulated OA environment was set up in a glovebox filled with $CO_2$ gas to mimic the elevated $CO_2$ in the atmosphere (Supplementary Fig. 1). A programmable $CO_2$ controller (ICC-500T, INKBIRD) was utilized to maintain a consistent atmospheric $CO_2$ concentration of 1000± 50 ppm. Four replicates of larvae were treated with ambient air and elevated $CO_2$ conditions, respectively. The larval samples were allocated to different groups for 24 h of treatment during the trochophore or 48 h treatment at later larval stages, to ensure adequate $CO_2$ dissolution. The number of larvae in each replicate was optimized based on the guidelines from the Mariculture Center (Supplementary Table 2). After a certain period of $CO_2$ exposure, the larvae from each treatment replicate were harvested in 7 ml TRIzol Reagent for RNA isolation.

### Evaluation of shell organic matrix synthesis by larval oysters under $CO_2$ exposure

Shell formation in 12-hour-old trochophores under the $CO_2$ exposure was observed after 24-h treatment. Early D-shape larvae were evaluated for the synthesis of shell organic matrix materials and calcification of the shell using fluorescent dyes calcein and calcofluor, respectively[70,76]. Calcein (MP Bio-medicals), a calcium-dependent fluorescent signal (UV channel, Exc: 408 nm/Em: 450–490 nm), was utilized for $CaCO_3$ staining of the calcified shell (final concentration 10 mM in 0.01% DMSO). Calcofluor White Stain (Sigma-Aldrich), a chitin-staining fluorescent signal (FITC channel, Exc: 488 nm/Em: 520–560 nm), was used to visualize the shell organic matrix (final concentration 0.02 mM in 0.01% DMSO). Before staining, overnight fixed larvae were washed with DPBS to remove the 4% PFA. Calcein was applied to the rinsed larvae for 2 h. After the DPBS rinsing to remove the excess Calcein dye, the larval samples were stained with the Calcofluor

White Stain. After 30 min of Calcofluor staining, the larvae were rinsed again with DPBS to remove Calcofluor and immediately imaged with a fluorescent microscope.

### CaM-inhibition assay for larval oyster shell development

W-7 was used to assess the function of the calcium-CaM signaling pathway in the biomineralization process during the early development of *C. virginica* larvae. Approximately 100 early D-shape larvae (24 h old) were separately treated with 2 mL of different concentrations of W-7 (1 μM, 2 μM, and 5 μM) diluted in filtered seawater for 24 h in the dark at room temperature. After treatment, 10 larvae per group were fixed in 4% PFA at 4 °C overnight for shell biogenesis analysis. The early shell development was assessed using Calcein and Calcofluor White Stain. The areas occupied by calcein and calcofluor on a single valve of each larva were manually outlined in ImageJ to determine the ratio of calcein to calcofluor as an indicator of shell biogenesis[70,76]. In addition, the fluorescence intensity of calcofluor in larval samples was quantified to estimate the production of shell matrix materials.

### Recovery experiment of CaN on regulating SMPs expression in *C. virginica* mantle cells under elevated $CO_2$

*C. virginica* mantle cells treated with CaN during elevated $CO_2$ exposure were used to investigate the involvement of CaN in regulating the expression of SMPs under simulated OA conditions. A 5000U human CaN stock solution (BML-SE163, Enzo), where 1U of calcineurin catalyzes the dephosphorylation of 1 mM substrate per minute, was mixed with 10 μM PEI (1:1 ratio) for 30 min incubation to facilitate cell delivery. The mixture was then diluted with oyster cell culture working medium to prepare 20U and 100U CaN-supplemented cell culture medium. On the 5th dpe, the mantle cells were exposed to 1.5% $CO_2$ exposure in CaN- supplemented media for 48 h. Treated cells were harvested for RNA isolation. The mantle cells from four *C. virginica* oysters were used in both the 20U and 100U CaN treatment groups.

### Quantification of gene expression in *C. virginica* mantle cells and *C. virginica* larvae

Total RNA was extracted from *C. virginica* mantle cells across different treatment groups using a standard Trizol-chloroform protocol[116], followed by DNA removal with the Ambion TURBO DNA-free Kit (Thermo Fisher Scientific). SuperScript IV Reverse Transcriptase (Invitrogen) was used to synthesize cDNA. The cDNA samples were used for SYBR Green qPCR analysis on a QuantStudio 3 Real-Time PCR System (Applied Biosystems). The qPCR program was run at 50 °C for 2 min, 94 °C for 2 min, followed by 40 cycles at 94 °C for 30 s, 55 °C for 30 s, and 72 °C for 30 s. The $2^{-\Delta\Delta Ct}$ method was used to analyze the gene expression data.

The mRNA expression levels of CaM, CaN, and four SMP-encoding genes, were measured under different treatment groups. Nucleotide sequence information of all the analyzed genes of *C. virginica* was collected from the NCBI RefSeq genome assembly C_virginica-3.0: GCF_002022765.2, including *Cv-CaM* (LOC111129443), *Cv-CaN* (LOC111120874), *Cv-Nacrein* (LOC111136431), *Cv-Pif97* (LOC111108062), *Cv-Tyr* (LOC111131635), and *Cv-Chits* (LOC111128079). Besides, the amino acid sequences used for multiple alignment analyses were retrieved from the NCBI protein database.

Candidate genes previously characterized from molluscan species, including *P. fucata*, *M. gigas*, and *Atrina rigida* were used to perform BLASTp analysis against the whole genome derived from *C. virginica* to identify the homologous mRNA sequences of *Cv-CaM* (MG029431), *Cv-CaN* (XM_022461887.1), *Cv-Pif97* (XM_022443581), *Cv-Nacrein* (XM_022487278), *Cv-Tyr* (XM_022479257.1), and *Cv-Chits* (XM_022473493) (Supplementary Table 3). The targeted genes for qPCR analysis were identified based on their amino acid similarity from the BLASTp results. The specificities of the primers designed for the qPCR analysis by Integrated DNA Technologies (IDT) were confirmed by running the amplicons on a 7% acrylamide TBE gel (Supplementary Fig. 2). All the

Ct values of the biomarkers were normalized by those of the housekeeping gene *Elongation factor 1α* gene (*Cv-ef1α*, BG624869.1)[34].

The Trizol-stored larval samples were homogenized using a microtube homogenizer (Bertin technologies). The homogenized samples were utilized for qPCR analysis to evaluate the relative expression levels of the biomineralization-related biomarkers (Supplementary Table 3) following the procedures for quantification of gene expression in *C. virginica* mantle cells.

### Statistics and reproducibility

For primary cell culture experiments, mantle cells were isolated from 4 to 8 independent oysters per treatment group, which were considered biological replicates. For each individual oyster, isolated mantle cells were cultured in 2–3 parallel culture flasks (technical replicates). Measurements from technical replicates were averaged for each oyster prior to statistical analysis.

For the *vivo* study, sample sizes for RNA extraction from oyster larvae were defined at the level of the culture beaker. Each beaker containing a stage-dependent number of larvae (Supplementary Table 2). Larval densities in each beaker were estimated using a hemacytometer. For RNA isolation, each 1 mL aliquot collected by TRIzol reagent was treated as a single measurement.

Comparisons of intracellular calcium fluorescent signal, gene expressions of calcium-signaling pathway related genes and SMPs at mRNA level, IF quantification of *Cv*-CaM and *Cv*-CaN, organic matrix/shell area ratio, Calcofluor fluorescent intensity, and CaN phosphatase activity were performed in R version 3.6.2[117,118]. A Student's *t* test was used for two-group comparisons, while a one-way ANOVA followed by Westfall's post hoc test was used for multiple comparisons versus the control group. For the comparison of long-term intracellular calcium fluorescent signal in mantle cells (Fig. 1H), one replicate was excluded prior to statistical analysis because its FITC-A value was remarkably below the range of the control and treatment groups and indistinguishable from the autofluorescence group (Supplementary Fig. 15), which indicated a technical failure in signal detection in flow cytometry. This exclusion was determined objectively by comparing the value against the ranges of the control and autofluorescence groups. All quantitative data were expressed as mean 95% confidence interval (CI). In all the statistical evaluations, $P < 0.05$ was considered statistically significant.

### Reporting summary

Further information on research design is available in the Nature Portfolio Reporting Summary linked to this article.

### Data availability

Data will be made available on request. Individual data points underlying the figures are provided in Supplementary Data 1. Original, uncropped gel electrophoresis images are provided in Supplementary Fig. 2b.

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

## Acknowledgements

We would like to thank Dr. Michael Wetz's group at the TAMU-Corpus Christi, Harte Research Institute for assisting us with the flow cytometry work. We would like to thank Dr. Nin Gan from TAMU-Corpus Christi for providing the oyster dissection image. This study was supported by the National Science Foundation (2046049 and 1903719).

## Author contributions

C.H. was responsible for conceptualization, experimental design, data analysis, and writing the original draft and revisions. J.M. and C.H. provided technical and material support and contributed to manuscript review. L.M. was responsible for supervision and manuscript review. in drafting and revision of the manuscript. W.X. was responsible for conceptualization, supervision, experiment design, manuscript review, and funding acquisition. All authors discussed the results and commented on the manuscript. All authors read and approved of the final manuscript.

## Competing interests

The authors declare no competing interests.
