## [Transparent Peer Review file · Communications Biology]

Ocean acidification disrupts the biomineralization process in the oyster *Crassostrea virginica* via intracellular calcium signaling dysregulation

Corresponding Author: Dr Wei Xu

Version 0:

Reviewer comments:

Reviewer #1

(Remarks to the Author)

This manuscript investigates the effects of ocean acidification (OA) on shell formation in the eastern oyster *Crassostrea virginica*. Using both in vitro mantle epithelial cell cultures and in vivo larval experiments, the authors demonstrated that elevated pCO₂ disrupts intracellular calcium homeostasis, leading to upregulation of calmodulin (CaM), downregulation of calcineurin (CaN), and abnormal expression of SMPs. These molecular alterations are linked to defects in shell structure and organic matrix deposition. Pharmacological inhibition of CaM reproduced the OA-induced abnormalities, while CaN supplementation restored normal SMP expression, suggesting that the CaM–CaN signaling pathway is a key mechanism underlying OA-induced shell deformities. The study provides new molecular insights into how OA interferes with biomineralization at the cellular signaling level.

To publish the paper, the authors must improve the whole manuscripts and data.

The authors must clearly show the genome, gene and amino acid database of *C. virginica* in the method.

The authors show the amino acid sequence of each gene from *C. virginica* and the alignments with other homologues from other species.

The schematic domain structure of each protein is also necessary.

It is difficult to know the domain structure and conserved sequence of each gene.

The authors must show the size of shell after treatments.

It is better to see the shell microstructure using SEM after the treatments.

It is difficult to understand the relationship between Ca-signaling and environmental pH.

1.5% CO₂ condition is too high for the estimation of environmental change. The readers want to see the dose dependent data for the concentration of CO₂, because the gene expression easily alter depending on many factors. If the gene expression change occurred as dose dependent manner, the validation of results will increase.

Reviewer #2

(Remarks to the Author)

Dear editor,

With great interest I read the manuscript (COMMSBIO-25-6928) by Huang and co-workers you asked me to review for Communications Biology. I applaud the thorough execution of various cell-physiological experiments that the authors present. The results are solid and well-described. However, I noticed important flaws in the methods and in the interpretation of the results. These need to be fixed before acceptance of this manuscript for publication. In short, the applied CO₂ concentration unlikely had the effect as was assumed by the authors. This could be mitigated by a more accurate description

of what actually happened when applying a higher CO₂ level. Secondly, the results cannot be generalized/ extended in the way the authors do: some of the conclusions are simply not supported by the results. Finally, the methods lack sufficient details to evaluate the results properly. Below, I will explain these three issues in more detail. In the annotated pdf, I have inserted a number of more technical and textual suggestions.

Sincerely,

Lennart de Nooijer

The applied CO₂ level.

If I understand correctly (Methods, line 521-530), concentrations were raised to 1.5% to test the effect of CO₂ on calcium signaling. This equals 15,000 ppm (current atmospheric concentrations are ~420 ppm). Such a high level far exceeds any projected future scenario (highest estimates are around 1000 ppm for the year 2100), so that the reported response of these oysters are not directly of practical importance. It may still be that the direction of physiological change under these conditions indicate what will happen in the near future, but the difference between control and 1.5% CO₂ may not be simply a multiple of the effect of 0.1% CO₂ compared to current atmospheric concentrations. On top of that, the change in CO₂ was applied instantaneously, with no time for any adaptation.

For a more robust analysis of the effect of carbon dioxide on cell physiological processes, a range of CO₂ levels must be applied, preferably including a lower-than-ambient CO₂ concentration to test whether the physiological effect does not depend on simply any change in CO₂ level. It is also necessary to monitor the carbonate chemistry of the medium to test the stability of the supposed carbon dioxide concentrations. The Methods does not mention any effort to prevent exchange of CO₂ between culture media and overlying air. If no specific measures were taken, the 1.5% would quickly diminish due to outgassing. It is also necessary to provide more details on the approach to reach the intended 1.5%: was a larger volume of medium bubbled with CO₂ and for how long? What was the effect on pH of the medium? Etc. A chemical characterization of the media is important as the extreme conditions aimed for here likely had a large impact on basic chemistry of the media. This is, amongst others, important for evaluating the results described in lines 201-215.

From Results to Conclusions

Unfortunately, only one (extremely) elevated carbon dioxide concentration was applied, which makes it impossible to reliably infer how intracellular Ca-regulation will behave at e.g. 800 or 1000 ppm CO₂. Moreover, the cells studied here are naturally not in direct contact with seawater, but separated by several layers of tissue (Figure 6G). The response to the extreme 1.5% CO₂ may easily lead to the opposite conclusion: with such a moderate physiological response it is likely that the (comparatively slow) changes in the coming decades will not severely impact intracellular calcium-regulation. The introduction (e.g. line 132), discussion (e.g. 330) and conclusions (line 497-498) mention 'OA conditions', but this is misleading.

Methods

Line 540-546: it is not completely clear in what order things were done. If I read correctly, cells were incubated with Fluo-4 AM for one hour at ambient pCO₂ and then, cells were exposed to the elevated CO₂ concentration. But were the cells still in the Fluo-4 AM when the CO₂ was increased? If so, the high-CO₂ treated cells also were incubated with Fluo-4AM for longer than the untreated cells. If not, the intensity of the Fluo-4 AM may well have suffered from bleaching/ leaking out of the cells. Or was there a control group in which the CO₂ remained the same, but that was compared to the cells under 1.5% CO₂?

Lines 547-555: were the cells still in the 1.5% CO₂ when the Fluo-4 AM was loaded into the cells? I guess not, because pH of the medium is known to affect Fluo-4 AM uptake into the cells. This is important since the long-term effect on intracellular Ca²⁺ concentrations is much higher than the short-term exposure. In addition: do the two different methods to measure the fluorescent intensity (fluorescence microscopy for the short, and flow cytometry for the long-term incubations) allow comparison? This result would have been more robust if (e.g.) the long-term incubated cells were also observed under the fluorescent microscope.

Lines 695-700. I don't see why expressing the variability of the calculated averages is done with an SE. It is a poor indicator for the reliability of the calculated average, especially given the low number of replicates (e.g. figure 2). Either explain or calculate confidence intervals to indicate between which values the calculated average likely falls.

Version 1:

Reviewer comments:

Reviewer #1

(Remarks to the Author)

The authors used the cDNA database to identify the SMP genes from the genome database. The authors must show the cDNA database in the manuscript.

Reviewer #2

(Remarks to the Author)

I have read the updated version of the manuscript of Huang and co-workers. They answered all of my initial comments in a satisfactory way, except for one. I see that the variability in the data is expressed as C.I. in the text, but not yet in the figures. In the latter, the variability is still depicted as SE, which is not justified by the relatively small amount of data per treatment. I

suggest to change this too.

RESPONSE TO REVIEWER COMMENTS

COMMSBIO-25-6928

Huang *et al.*

“Ocean acidification disrupts the biomineralization process in the oyster *Crassostrea virginica* via intracellular calcium signaling dysregulation”

We would like to thank the editor and the reviewers for their thoughtful comments, constructive suggestions, and careful evaluation of our manuscript. In response to the editor’s request for the revised submission, (1) individual data points have been used to replace all bar graphs, and (2) all numerical source data for the figures and charts have been provided in an Excel file as supplementary material.

Below, we provide detailed, point-by-point responses to the comments and suggestions from the reviewers.

Reviewer #1:

1. The authors must clearly show the genome, gene and amino acid database of *C. virginica* in the method.

Response:

We have added the detailed information on the *C. virginica* genome (NCBI RefSeq genome assembly *C_virginica*-3.0: GCF_002022765.2), gene (*Cv-CaM*: LOC111129443, *Cv-CaN*: LOC111120874; *Cv-Nacrein*: LOC111136431; *Cv-Pif97*: LOC111108062; *Cv-Tyr*: LOC111131635; *Cv-Chits*: LOC111128079), and amino acid data (NCBI protein database, *Cv-CaM*: ATW76006.1; *Cv-CaN*: XP_022317595). The detailed revision is highlighted in method section 5.11 (**Lines: 741-746**).

2. The authors show the amino acid sequence of each gene from *C. virginica* and the alignments with other homologues from other species.

Response:

For the protein calmodulin and calcineurin of *C. virginica*, we showed the multiple alignment with the identified proteins from *Pinctada fucata* (bivalve), *Haliotis discus* (gastropod), *Drosophila melanogaster* (arthropod), and *Homo sapiens* (vertebrate). The conserved amino acid sequence suggested a potential conserved function of the calcium/calmodulin/calcineurin signaling pathway. The detailed revision is highlighted in the discussion section (**Lines 353-357**) and is included in the supplementary data (**Figures S8 and S9**).

Despite species-specific variations in shell matrix proteins (SMPs) composition, SMP repertoires have been reported to share four functional domains, including von Willebrand factor type A domain (VWA), chitin-binding-2 domain (CB-2), carbonic anhydrase domain (CA), and tyrosinase domain in diverse molluscan species^{1,2}. To

evaluate shell construction at the molecular level, four *C. virginica* SMPs containing one of those conserved domains, including Cv-nacrein, Cv-Pif97, Cv-Tyr, and Cv-Chits, were aligned with homologous proteins identified in three bivalve species (*Pinctada fucata*, *Magallana gigas*, and *Mytilus edulis*). This analysis revealed the conserved domains, namely VWA, CA, CB-2, and Tyrosinase, shared by bivalve shell matrix proteins. The detailed revisions are highlighted in the results section (Lines 201-203) and are presented in the Supplementary Data (Figures S10–S13) to validate the reasons for using these four encoded genes as shell formation indicators at the transcription level.

3. The schematic domain structure of each protein is also necessary. It is difficult to know the domain structure and conserved sequence of each gene.

Response:

We agree that it is important to illustrate the domain locations, structural features, and conserved amino acid sequences of each analyzed gene. The locations of schematic domains and the predicted 3D structure for each gene analyzed in this study are included in the Supplementary Data (Figures S8-S13).

4. The authors must show the size of shell after treatments. It is better to see the shell microstructure using SEM after the treatments.

Response:

Previous studies have demonstrated that OA simulations can lead to bivalve shell dissolution and disruption of shell microstructure as observed under SEM³⁻⁶. Our study primarily focused on the physiological responses of bivalves to OA conditions and the regulatory mechanisms associated with shell morphology patterns. We compared the size of oyster shells using shell surface area as an indicator in 48-hour D-shape and 20-day-old veliger juveniles. There were no significant changes in the shell surface area under different treatments, except the D-shape larvae under 5 μ M W-7 exposure showed significantly smaller shell surface area than the control group (Supplementary Data, Figure S5). The detailed revision and explanation are highlighted in the discussion section (Lines 461-476 and 490-494) and in the supplementary data (Figure S5).

5. It is difficult to understand the relationship between Ca-signaling and environmental pH.

Response:

From a chemical perspective, shell disruption under OA can be explained by two major mechanisms. First, atmospheric CO₂ absorbed by seawater releases H⁺ and competes

with CaCO₃ nucleation for CO₃²⁻, leading to reduced CO₃²⁻ availability for the calcification process ⁷. The relevant chemical reactions are:

Second, the acidic environment also dissolves the existing CaCO₃ in the shell structure, resulting in fragile shell structures ⁸. Here is a potential chemical reaction formula:

Changes in pH in the extracellular environment alter proton gradients and cell membrane potential in oyster mantle epithelial cells ⁹. Fluctuations in the carbonate chemistry system and extracellular Ca²⁺ concentrations under OA may also be associated with Ca²⁺ channel activity and disrupt intracellular Ca²⁺ levels, which have been reported in oyster hemocytes ^{10,11}. In this study, we aimed to explore whether the Ca²⁺ signaling pathway, which is a classical pathway in vertebrate biomineralization ^{12,13}, such as bone and fin formation, affects the biomineralization process of marine bivalve shell formation and is impacted by OA.

Overall, OA is a synergistic change in the marine environment, rather than just the reduction in environmental pH. It involves changes in the whole carbonate chemistry system in both seawater and bivalve extrapallial fluid, where the mantle epithelial cells are directly exposed (Figure 6). We have clarified this relationship in the introduction section, and the detailed revision is highlighted (Lines: 115, 116-117, 118-126, and 134-138).

6. 1.5% CO₂ condition is too high for the estimation of environmental change. The readers want to see the dose dependent data for the concentration of CO₂, because the gene expression easily alters depending on many factors. If the gene expression change occurred as dose dependent manner, the validation of results will increase.

Response:

Our lab's previous work has acknowledged the importance of dose-dependent validation and has demonstrated the expression trend of four calcium-binding proteins, including calmodulin (CaM), in oyster mantle cells exposed to different CO₂ concentrations (1% and 2.5%), suggesting that the transcriptional responses of mantle cells to CO₂ are dose-dependent ¹⁴.

The mantle cells were cultured in an artificial medium containing critical components to support the cells *in vitro*. These components could make the culture medium markedly different from the natural seawater. For example, the pH of the cell culture medium under ambient CO₂ level is about 7.8, which would be considered an acidified condition in natural seawater. The buffering capability of the cell culture medium is much greater than that of natural seawater. 1.5% CO₂ input to the medium is equivalent to changing

the $p\text{CO}_2$ level from 400 (ambient condition) to 1000 (atmospheric CO_2 levels in year 2100) ppm, which decreases the pH by 0.4 (Supplementary Data, Figure S6). Using environmental-like CO_2 concentration (e.g., 0.1% CO_2) would likely fail to reflect naturally closed cellular or molecular responses under controlled *in vitro* conditions. We have clarified the reasons for using 1.5% CO_2 treatment in the discussion section with more details (Lines: 369-383).

Reviewer #2:

1. If I understand correctly (Methods, line 521-530), concentrations were raised to 1.5% to test the effect of CO_2 on calcium signaling. This equals 15,000 ppm (current atmospheric concentrations are ~420 ppm). Such a high level far exceeds any projected future scenario (highest estimates are around 1000 ppm for the year 2100), so that the reported response of these oysters is not directly of practical importance.

On top of that, the change in CO_2 was applied instantaneously, with no time for any adaptation. For a more robust analysis of the effect of carbon dioxide on cell physiological processes, a range of CO_2 levels must be applied, preferably including a lower-than-ambient CO_2 concentration to test whether the physiological effect does not depend on simply any change in CO_2 level. It is also necessary to monitor the carbonate chemistry of the medium to test the stability of the supposed carbon dioxide concentrations.

Response:

The reason for applying this high level of CO_2 was to trigger measurable cellular responses *in vitro* and explore the molecular and signaling mechanisms of bivalve shell formation under OA conditions.

As described in our response to Reviewer 1's comment, because of the different properties of cell culture media and natural seawater, changing the CO_2 level from ambient level to 1.5% for cell culture media is equivalent to increasing the atmospheric CO_2 level from 400 to 1000 ppm in the natural environment, resulting in a drop of pH by 0.4.

In our previous study, we conducted an experiment with different concentrations of CO_2 exposure (1% and 2.5%) and found a trend of transcriptional responses in the oyster mantle cells to elevated CO_2 ¹⁴. Based on these preliminary results, a 1.5% CO_2 treatment was selected as an appropriate exposure level to stimulate measurable cellular and molecular responses *in vitro* in the mantle epithelial cells

In addition, we estimated the carbonate chemistry parameters, including $[\text{HCO}_3^-]$ and $[\text{CO}_3^{2-}]$, in the cell culture medium under CO_2 exposure based on the pH and $p\text{CO}_2$

(Supplementary Data, Table S4). It is surprising to see that the ranges of these parameters are similar to the parameters detected in the bivalve extrapallial fluid, where the mantle epithelial cell lives in bivalves, under the prospective CO₂ level (2050 ppm)¹⁵. Therefore, although the experimental CO₂ level appeared high, the actual cellular exposure in terms of pH and carbonate chemistry is within the physiological range relevant to OA studies.

The detailed revision and explanation are highlighted in the introduction section (Lines 118-126) and discussion section (Lines 369-383). The calculated carbonate chemistry parameters are added to the supplementary data (Table S4).

2. The Methods does not mention any effort to prevent exchange of CO₂ between culture media and overlying air. If no specific measures were taken, the 1.5% would quickly diminish due to outgassing. It is also necessary to provide more details on the approach to reach the intended 1.5%.

Response:

The CO₂ concentration during mantle epithelial cell culture was regulated using a CO₂-controlled incubator (VWR Water-Jacked CO₂ Incubator–Model 2325). The incubator has an automatic gas regulator and a CO₂ sensor, ensuring a stable atmospheric concentration of 1.5% CO₂ throughout the entire exposure period.

The detailed revision and reclarification are highlighted in the method section 5.2 (Lines 578-581).

3. Was a larger volume of medium bubbled with CO₂ and for how long? What was the effect on pH of the medium? Etc. A chemical characterization of the media is important as the extreme conditions aimed for here likely had a large impact on basic chemistry of the media. This is, amongst others, important for evaluating the results described in lines 201-215.

Response:

Oyster larvae were exposed to an atmosphere of 1000 ppm CO₂, as predicted by the 2021 IPCC report ¹⁶, within a sealed glovebox system. Rather than bubbling CO₂ directly into the seawater to alter the pH, our setup more realistically simulated air-water CO₂ exchange at the water surface (Supplementary Data, Figures S1A and B). The larvae were cultured in 700mL seawater contained in 1 L plastic beakers inside the glovebox. After 48 hours of exposure to 1000 ppm atmospheric CO₂, the average seawater pH decreased significantly from 7.85 to 7.30 (Supplementary Data, Figure S1D), validating the effectiveness of the OA simulation system in our study. The calculated [CO₃²⁻] and saturation states of aragonite and calcite in 1000 ppm CO₂-treated seawater in our study also showed similarly reduced ranges compared to the previous acidification seawater treatment study (Supplementary Data, Table S4) ^{15,17}.

These results suggest that the observed shell matrix deformities and abnormal SMP transcription levels may be associated with the changes in the carbonate chemistry system under mimicked acidified conditions.

The detailed revision is highlighted in the result section (Lines 214-219) and the discussion section (Lines 461-476). The calculated carbonate chemistry in CO₂ treated seawater is added to the supplementary data (Table S4).

4. Unfortunately, only one (extremely) elevated carbon dioxide concentration was applied, which makes it impossible to reliably infer how intracellular Ca-regulation will behave at e.g. 800 or 1000 ppm CO₂. Moreover, the cells studied here are naturally not in direct contact with seawater, but separated by several layers of tissue (Figure 6G). The response to the extreme 1.5% CO₂ may easily lead to the opposite conclusion: with such a moderate physiological response it is likely that the (comparatively slow) changes in the coming decades will not severely impact intracellular calcium-regulation. The introduction (e.g. line 132), discussion (e.g. 330) and conclusions (line 497-498) mention 'OA conditions', but this is misleading.

Response:

In this study, we used both an *in vitro* mantle epithelial cell model and an *in vivo* oyster larval model to study the effects of ocean acidification induced by elevated *p*CO₂ on bivalve shell formation and its underlying molecular mechanisms. As described earlier, the 1.5% CO₂ exposure in the *in vitro* experiment resulted in only a 0.4 decrease in the culture medium pH. Given the specifically required extracellular conditions and pH buffering system optimized to maintain mantle epithelial cell viability for cellular functional and response analyses, the *in vitro* system has to be designed differently from a natural-like environment, providing a defined system to study the cellular responses to elevated *p*CO₂. Therefore, we incorporated an *in vivo* study using oyster larvae to validate the role of the calcium signaling pathway in regulating bivalve shell formation, which we had identified in the *in vitro* study. The *in vivo* experiment exposed oyster larvae to a projected elevated atmospheric CO₂ level (1000 ppm) and exhibited patterns consistent with the *in vitro* results, including altered expression of calcium signaling pathways and SMPs. The *in vivo* larval model also showed abnormal production and arrangement of the shell organic matrix.

We have rephrased the wording, replacing "OA condition" with "OA induced by elevated *p*CO₂" to more accurately display the results from mantle epithelial cells and clarify the conclusions drawn from the *in vitro* study. The detailed revision is highlighted in the introduction section (Lines 143 and 149), discussion section (Line 346), and conclusion section (Line 552).

5. Line 540-546: it is not completely clear in what order things were done. If I read correctly, cells were incubated with Fluo-4 AM for one hour at ambient pCO₂ and then, cells were exposed to the elevated CO₂ concentration. But were the cells still in the Fluo-4 AM when the CO₂ was increased? If so, the high-CO₂ treated cells also were incubated with Fluo-4AM for longer than the untreated cells. If not, the intensity of the Fluo-4 AM may well have suffered from bleaching/ leaking out of the cells. Or was there a control group in which the CO₂ remained the same, but that was compared to the cells under 1.5% CO₂?

Response:

For the short-term CO₂ exposure experiment, both control and CO₂-treated mantle cell samples were stained with Fluo-4 AM for 1 hour. After staining, the Fluo-4 AM-containing medium was removed, and the cells were cultured in fresh medium for 1 hour under either ambient air or 1.5% CO₂ exposure.

For the concern about the leaking or photobleaching of calcium dye, the Fluo-4 AM is specifically designed to measure intracellular calcium. The AM group (acetoxymethyl ester) allows the dye to cross the cell membrane. The Fluo-4 AM does not bind extracellular calcium and remains non-fluorescent until the AM group is hydrolyzed by intracellular esterases, converting it to Fluo-4, which can bind intracellular calcium and emit fluorescent signals. Once cleaved, Fluo-4 is charged and largely retained in the cytoplasm. Under typical experimental conditions, Fluo-4 remains intracellularly for a few hours, which is sufficient for short-term live cell Ca²⁺ imaging. Besides, because the cells were cultured in a dark incubator, photobleaching is negligible.

The detailed clarification is highlighted in the method section (Lines 592-594 and 598-600).

6. Lines 547-555: were the cells still in the 1.5% CO₂ when the Fluo-4 AM was loaded into the cells? I guess not, because pH of the medium is known to affect Fluo-4 AM uptake into the cells. This is important since the long-term effect on intracellular Ca²⁺ concentrations is much higher than the short-term exposure. In addition: do the two different methods to measure the fluorescent intensity (fluorescence microscopy for the short, and flow cytometry for the long-term incubations) allow comparison? This result would have been more robust if (e.g.) the long-term incubated cells were also observed under the fluorescent microscope.

Response:

The isolated cells were stained with Fluo-4 AM for 1 hour and then rinsed before 24-hour CO₂ exposure, ensuring that the cells were not exposed to 1.5% CO₂ during the Fluo-4 AM staining process. This ensures that dye loading was consistent between control and CO₂-treated cells, avoiding the potential impact of medium pH on Fluo-4 AM staining.

Although Fluo-4 is largely retained in the cytoplasm for several hours, a small amount of the dye can still gradually leak out over time through passive diffusion. Continuous live imaging of the same cells over 24 hours without dye replenishment, as dye loading could be impacted by changing pH, would lead to faint signals, making measurements under a fluorescent microscope unreliable. To overcome this, we used flow cytometry to analyze cells from different patches after 24 hours of culture under either ambient air or CO₂ treatment. While this method did not allow long-term live cell imaging, it could also provide a robust comparison of relative fluorescence between control and CO₂-treated groups. Ideally, after 24 hours of incubation, mantle cells with a higher intracellular calcium level will exhibit more dye and retain more fluorescent signals intracellularly. It enables a reliable assessment of changes in intracellular calcium concentration after 24 hours of different treatments. The detailed clarification is highlighted in the method section (Lines 607-608 and 6135-615).

7. Lines 695-700. I don't see why expressing the variability of the calculated averages is done with an SE. It is a poor indicator for the reliability of the calculated average, especially given the low number of replicates (e.g. figure 2). Either explain or calculate confidence intervals to indicate between which values the calculated average likely falls.

Response:

We agree that with a low number of replicates, CI is more informative to demonstrate data variability. 95% confidence interval has been calculated and indicated on all relevant figure plots. The revisions are highlighted in yellow in the method section (Line 767) and the result sections (Lines: 181, 184, 201-206, 240-242, 245, 249-251, 253, 255-256, 271-272, 274-276, 279, 281, 284-285).

8. Original version line 121: consider replacing by 'disrupting' or something similar

Response:

The word "affecting" has been replaced with "disrupting" (Line 128).

9. Original version lines 137-142: This is already in Results and Discussion, perhaps better remove here.

Response:

The repeated summary of the results has been removed from the introduction section.

10. Original version lines 264: I wonder a bit about the number of digits for these numbers: does the method to detect (e.g.) CaM allow for such a precise analysis of the difference in expression?

Response:

We have calculated the 95% CI in the difference in protein expression level and added it to lines 249 and 281.

REFERENCE:

1. Arivalagan, J. *et al.* Insights from the Shell Proteome: Biomineralization to Adaptation. *Molecular Biology and Evolution* **34**, 66–77 (2017).
2. Zhao, R. *et al.* Dual Gene Repertoires for Larval and Adult Shells Reveal Molecules Essential for Molluscan Shell Formation. *Molecular Biology and Evolution* **35**, 2751–2761 (2018).
3. Talmage, S. C. & Gobler, C. J. Effects of past, present, and future ocean carbon dioxide concentrations on the growth and survival of larval shellfish. *Proceedings of the National Academy of Sciences* **107**, 17246–17251 (2010).
4. Kapsenberg, L. *et al.* Ocean pH fluctuations affect mussel larvae at key developmental transitions. *Proceedings of the Royal Society B: Biological Sciences* **285**, 20182381 (2018).
5. Zhang, Y. *et al.* The Inhibition of Ocean Acidification on the Formation of Oyster Calcified Shell by Regulating the Expression of Cgchs1 and Cgchit4. *Frontiers in Physiology* **10**, (2019).
6. Chandra Rajan, K. *et al.* Directional fabrication and dissolution of larval and juvenile oyster shells under ocean acidification. *Proceedings of the Royal Society B: Biological Sciences* **290**, 20221216 (2023).

7. Barker, S. & Ridgwell, A. Ocean acidification. *Nature Education Knowledge* **3**, (2012).
8. Feely, R. A. *et al.* Impact of anthropogenic CO₂ on the CaCO₃ system in the oceans. *Science* **305**, 362–366 (2004).
9. Ramesh, K., Hu, M. Y., Melzner, F., Bleich, M. & Himmerkus, N. Intracellular pH regulation in mantle epithelial cells of the Pacific oyster, *Crassostrea gigas*. *J Comp Physiol B* **190**, 691–700 (2020).
10. Wang, X. *et al.* Transcriptional changes of Pacific oyster *Crassostrea gigas* reveal essential role of calcium signal pathway in response to CO₂-driven acidification. *Science of The Total Environment* **741**, 140177 (2020).
11. Wang, X., Li, C., Lv, Z., Zhang, Z. & Qiu, L. A calcification-related calmodulin-like protein in the oyster *Crassostrea gigas* mediates the enhanced calcium deposition induced by CO₂ exposure. *Science of The Total Environment* **833**, 155114 (2022).
12. Cao, Z. *et al.* Calcineurin controls proximodistal blastema polarity in zebrafish fin regeneration. *Proceedings of the National Academy of Sciences* **118**, e2009539118 (2021).
13. Leser, J. M. *et al.* Osteoblast-lineage calcium/calmodulin-dependent kinase 2 delta and gamma regulates bone mass and quality. *Proceedings of the National Academy of Sciences* **120**, e2304492120 (2023).
14. Richards, M., Xu, W., Mallozzi, A., Errera, R. M. & Supan, J. Production of Calcium-Binding Proteins in *Crassostrea virginica* in Response to Increased Environmental CO₂ Concentration. *Frontiers in Marine Science* **5**, (2018).

15. Cameron, L. P., Grabowski, J. H. & Ries, J. B. Effects of elevated pCO₂ and temperature on the calcification rate, survival, extrapallial fluid chemistry, and respiration of the Atlantic Sea scallop *Placopecten magellanicus*. *Limnology and Oceanography* **67**, 1670–1686 (2022).
16. Masson-Delmotte, V. *et al.* Climate change 2021: the physical science basis. *Contribution of working group I to the sixth assessment report of the intergovernmental panel on climate change* **2**, 2391 (2021).
17. Fanguie, N. A. *et al.* A laboratory-based, experimental system for the study of ocean acidification effects on marine invertebrate larvae. *Limnology and Oceanography: Methods* **8**, 441–452 (2010).

RESPONSE TO REVIEWER COMMENTS

COMMSBIO-25-6928-A

Huang et al.

“Ocean acidification disrupts the biomineralization process in the oyster *Crassostrea virginica* via intracellular calcium signaling dysregulation”

We thank the editors and the reviewers for their time and effort in re-evaluating our 1st revised manuscript. Below, we provide detailed responses to the comments and suggestions from Reviewer 1 and Reviewer 2. All revisions made in response to the reviewers' comments and the editorial comments and requests are highlighted in yellow in the revised manuscript.

Reviewer #1:

#1 Comment

The authors used the cDNA database to identify the SMP genes from the genome database. The authors must show the cDNA database in the manuscript.

#1 Response

Thank you for Reviewer 1's suggestion. All SMP genes and calcium signaling pathway-related genes were all identified using the NCBI mRNA database for primer design. The corresponding NCBI mRNA accession number of each gene are as follows: *Cv-elf1* (BG624869.1), *Cv-CaM* (MG029431), *Cv-CaN* (XM_022461887.1), *Cv-Nacrein* (XM_022487278), *Cv-Pif97* (XM_022443581), *Cv-Tyr* (XM_022479257.1), and *Cv-Chits* (XM_022473493). These accession numbers have been added in the Methods section (Lines: 744-747) and in supplementary data Table S3.

Reviewer #2:

#1 Comment

I have read the updated version of the manuscript of Huang and co-workers. They answered all of my initial comments in a satisfactory way, except for one. I see that the variability in the data is expressed as C.I. in the text, but not yet in the figures. In the latter, the variability is still depicted as SE, which is not justified by the relatively small amount of data per treatment. I suggest to change this too.

#1 Response

Thank you for Reviewer 2's suggestion. We would like to clarify that, in the first submitted revision, most error bars in the figures had already been corrected to represent 95% confidence intervals. Only the error bars in Figure 1F and Figure 5 in the main text, as well as Supplementary Figures S1D, S6, and S7, were not updated at that time. Besides, the corresponding figure captions were also not revised, which might have caused confusion for the reviewers and editors. In the latest revision, all figures and captions have now been updated to display 95% confidence intervals.